# Endogenous formaldehyde is a memory-related molecule in mice and humans

Li Ai[1,11], Tao Tan [2,3,11], Yonghe Tang[4,11], Jun Yang[5,6], Dehua Cui[5,6], Rui Wang[5,6], Aibo Wang[5,6], Xuechao Fei[1], Yalan Di[1], Xiaoming Wang[1], Yan Yu[7], Shengjie Zhao[7], Weishan Wang[4], Shangying Bai[1,8], Xu Yang [9], Rongqiao He[1,10], Weiying Lin[4,12], Hongbin Han[5,6,12], Xiang Cai[8,12] & Zhiqian Tong[1]*

Gaseous formaldehyde is an organic small molecule formed in the early stages of earth's evolution. Although toxic in high concentrations, formaldehyde plays an important role in cellular metabolism and, unexpectedly, is found even in the healthy brain. However, its pathophysiological functions in the brain are unknown. Here, we report that under physiological conditions, spatial learning activity elicits rapid formaldehyde generation from mitochondrial sarcosine dehydrogenase (SARDH). We find that elevated formaldehyde levels facilitate spatial memory formation by enhancing N-methyl-D-aspartate (NMDA) currents via the C232 residue of the NMDA receptor, but that high formaldehyde concentrations gradually inactivate the receptor by cross-linking NR1 subunits to NR2B. We also report that in mice with aldehyde dehydrogenase-2 (*ALDH2*) knockout, formaldehyde accumulation due to hypofunctional ALDH2 impairs memory, consistent with observations of Alzheimer's disease patients. We also find that formaldehyde deficiency caused by mutation of the mitochondrial *SARDH* gene in children with sarcosinemia or in mice with *Sardh* deletion leads to cognitive deficits. Hence, we conclude that endogenous formaldehyde regulates learning and memory via the NMDA receptor.

[1] Alzheimer's Disease Center, Beijing Institute for Brain Disorders; Center for Brain Disorders Research, Capital Medical University, Beijing 100069, China. [2] Department of Physiology and Biophysics School of Medicine & Biomedical Sciences, State University of New York at Buffalo, Buffalo, NY 14214, USA. [3] Sichuan Provincial Hospital for Women and Children, Chengdu 610000, China. [4] Institute of Fluorescent Probes for Biological Imaging, School of Chemistry and Chemical Engineering, School of Materials Science and Engineering, University of Jinan, Shandong 250022, China. [5] Department of Radiology, Peking University Third Hospital, Beijing, China. [6] Key Laboratory of Magnetic Resonance Imaging Equipment and Technique, Beijing 100191, China. [7] Beijing Boai Hospital, China Rehabilitation Research Center, Beijing 100068, China. [8] Department of Physiology, Southern Illinois University School of Medicine, Carbondale, IL 62901, USA. [9] Section of Environmental Biomedicine, Hubei Key Laboratory of Genetic Regulation and Integrative Biology, College of Life Sciences, Central China Normal University, Wuhan 430079, China. [10] State Key Laboratory of Brain & Cognitive Science, Institute of Biophysics, CAS Key Laboratory of Mental Health, University of Chinese Academy of Sciences (UCAS), Beijing 100101, China. [11]These authors contributed equally to this work: Li Ai, Tao Tan, Yonghe Tang. [12]These authors jointly supervised this work: Weiying Lin, Hongbin Han, Xiang Cai. *email: tzqbeida@ccmu.edu.cn

Gaseous formaldehyde (MW = 30) is one of the earliest organic molecules containing C, H, and O elements in the earth's early evolution[1,2]. Actually, formaldehyde is the primary precursor for most complex organic compounds, including amino acids, RNA, DNA, and proteins[3]. It is notoriously known as an indoor air pollutant that induces memory deficits in animals[4], and cognitive decline in humans[5]. Formaldehyde is also present in every vertebrate cell as a possible byproduct of several metabolic reactions (e.g., methanol oxidation, DNA or histone demethylation)[6]. Further, relatively high formaldehyde concentrations (~300 μM, which is 41,000-fold lower than 37% Formalin contain: $1.23 \times 10^7$ μM formaldehyde) have been detected in the brains of healthy adult mice, rats, and humans[7,8], but it is unclear whether this is endogenous or from environmental contamination. Several recent studies suggested that formaldehyde may partially affect the activities of the N-methyl-D-aspartate (NMDA)-type glutamate receptor (NMDA-R, consisting of the NR1 subunit and an NR2 or NR3 subunit)[9,10]. Notably, NMDA-R is widely believed to be critical for learning and memory by triggering associative synaptic plasticity[11], suggesting a possible mechanism for formaldehyde-dependent cognitive modulations. However, whether endogenous formaldehyde affects cognitive activity through this or other pathways remains largely unknown.

Alzheimer's disease (AD) is a common neurodegenerative disease in the elderly characterized by persistent cognitive decline that has also been linked to disrupted formaldehyde metabolism[8,12]. Polymorphism in mitochondrial aldehyde dehydrogenase 2 (ALDH2), which degrades formaldehyde, is a major risk factor for late-onset AD in the Chinese population[13]. In fact, ALDH2 mutation in type-II diabetic patients is closely related to cognitive decline[14,15]. Injection of formaldehyde at pathological concentration (over 300 μM) indeed directly induces spatial memory deficits in healthy adult mice[7,8]. These findings suggest that ALDH2 mutation-related endogenous formaldehyde overload may contribute to cognitive disorders in AD.

Sarcosinemia is a rare pediatric neurodegenerative disease characterized by high levels of sarcosine in the blood and urine[16], mental retardation (low intelligence quotient, intelligence quotient), speech disorder, and ataxia[17]. It is a recessive inherited disease linked to loss-of-function mutations in the sarcosine dehydrogenase gene (SARDH), resulting in ~4-fold reduced SARDH activity in the mitochondria. SARDH produces free formaldehyde in the conversion of sarcosine to glycine[18], raising the possibility that the cognitive deficits of sarcosinemia are associated with brain formaldehyde deficiency.

In this study, we addressed the relationship between formaldehyde metabolism and cognitive disorders in children with sarcosinemia, AD dementia patients, and mice deficient in either SARDH ($Sardh^{-/-}$) or ALDH2 ($Aldh2^{-/-}$). The possible molecular mechanisms of formaldehyde-regulated NMDA-R activity via NR1 and/or NR2B subunits were also investigated. Our findings suggest that endogenous formaldehyde bidirectionally modulates cognition via NMDA-R, with both insufficiency and overabundance resulting in cognitive deficits.

## Results

### Learning activity induces rapid generation of formaldehyde.

To investigate the role of brain formaldehyde in cognitive processes, we examined whether hippocampal formaldehyde concentration changes after spatial learning activity in Morris water maze (MWM) in the wild-type adult male Sprague–Dawley (SD) rats. First, we observed that healthy rats exhibited a rapid reduction in escape latency during training and significantly greater time spent in the target quadrant compared to nontarget quadrants during the probe test (Fig. 1a, b), suggesting that a classical spatial memory was forming in these rats. Notably, this spatial learning was associated with a marked increase in hippocampal formaldehyde as detected using a free formaldehyde-sensitive fluorescent probe NaFA[19] (Fig. 1c). The magnitude of this elevation was ~30 μM on day 1 and ~50 μM on day 6 as quantified by high-performance liquid chromatography using a fluorescence detector (Fluo-HPLC)[20] (Fig. 1d). These data strongly suggest that hippocampus-related spatial learning activity elicits an elevation in hippocampal formaldehyde levels.

Long-term potentiation (LTP) is considered a critical synaptoplastic mechanism underlying hippocampal learning and memory[21], so we examined the changes in formaldehyde levels during hippocampal LTP induction in healthy rats. Late-phase LTP was induced by stimulating the Schaffer collateral pathway with three trains of high-frequency electrical stimulation (HFS, 100 Hz, 20 pulses) at 5-min intervals and recorded field excitatory postsynaptic potentials (fEPSPs) in the CA1 stratum radiatum in vivo (Fig. 1e). L-LTP was maintained for at least 2 h (fEPSP peak at 35 min after HFS (fold of baseline): $1.97 \pm 0.09$; fEPSP peak at 150 min after HFS: $1.29 \pm 0.04$; $n = 10$, $p < 0.01$) (Fig. 1f). Using the formaldehyde-sensitive fluorescent probe- NaFA to image formaldehyde intensity and Fluo-HPLC to quantify formaldehyde concentrations, we detected a marked increase (~50 μM) in hippocampal formaldehyde at 5 min post-HFS, which decreased over 1 h (Fig. 1g, h). These data indicate that formaldehyde generation mainly contributes to LTP induction.

Further, we investigated whether it is the excited hippocampal neurons that produce formaldehyde. One possible pathway is formaldehyde generation from the mitochondrial SARDH in the process of converting sarcosine to glycine[18,22], while excess formaldehyde can be degraded by ALDH2[23] (Fig. 1i). Our results of using double-staining fluorescence histochemistry showed that SARDH was mainly colocalized with the mitochondrial marker—Cox IV (Fig. 1j). To further identify whether mitochondrial SARDH in the cultured hippocampal neurons contributes to formaldehyde generation, the competitive SARDH inhibitor- methoxyacetic acid was incubated before the excitatory neurotransmitter- glutamate (Glu) was added into the cultured medium. The results showed that Glu induced a marked decline in sarcosine levels concomitant with the rapid formaldehyde elevation ($\Delta \approx 30$ μM). However, the mitochondrial SARDH inhibitor, methoxyacetic acid, suppressed sarcosine reduction and formaldehyde generation induced by Glu stimulation in vitro (Fig. 1k, l). These data confirm that this active formaldehyde generation is derived from mitochondrial SARDH.

To achieve real-time imaging this active formaldehyde generation in the synapse and soma of hippocampal neurons during the process of excitatory stimulation, a mitochondria-sensitive formaldehyde fluorescence probe (mito-FA-probe, this probe is sensitive to both formaldehyde in the mitochondrial and mitochondrial membranes[22]) was added into the stimulating and recording electrodes which were patched in the soma of neurons (Fig. 1m). Then we recorded multiple action potentials in the postsynaptic neurons after given a train with 20 pulses of electrical stimulation in the presynaptic neurons in the cultured hippocampal neurons (Fig. 1m). The results showed that typical multiple action potentials were formed in postsynaptic neurons associated with a transient increase and then a rapid decrease in the fluorescence intensity of mito-FA-probe in the axons, especially in the synapses and somas of the pre- and postsynaptic neurons (Fig. 1n, o), indicating that electrical stimulation directly induces endogenous formaldehyde generation from the mitochondria and this elevated formaldehyde rapidly infuses into synaptic clefts from the activated mitochondria. The synapses,

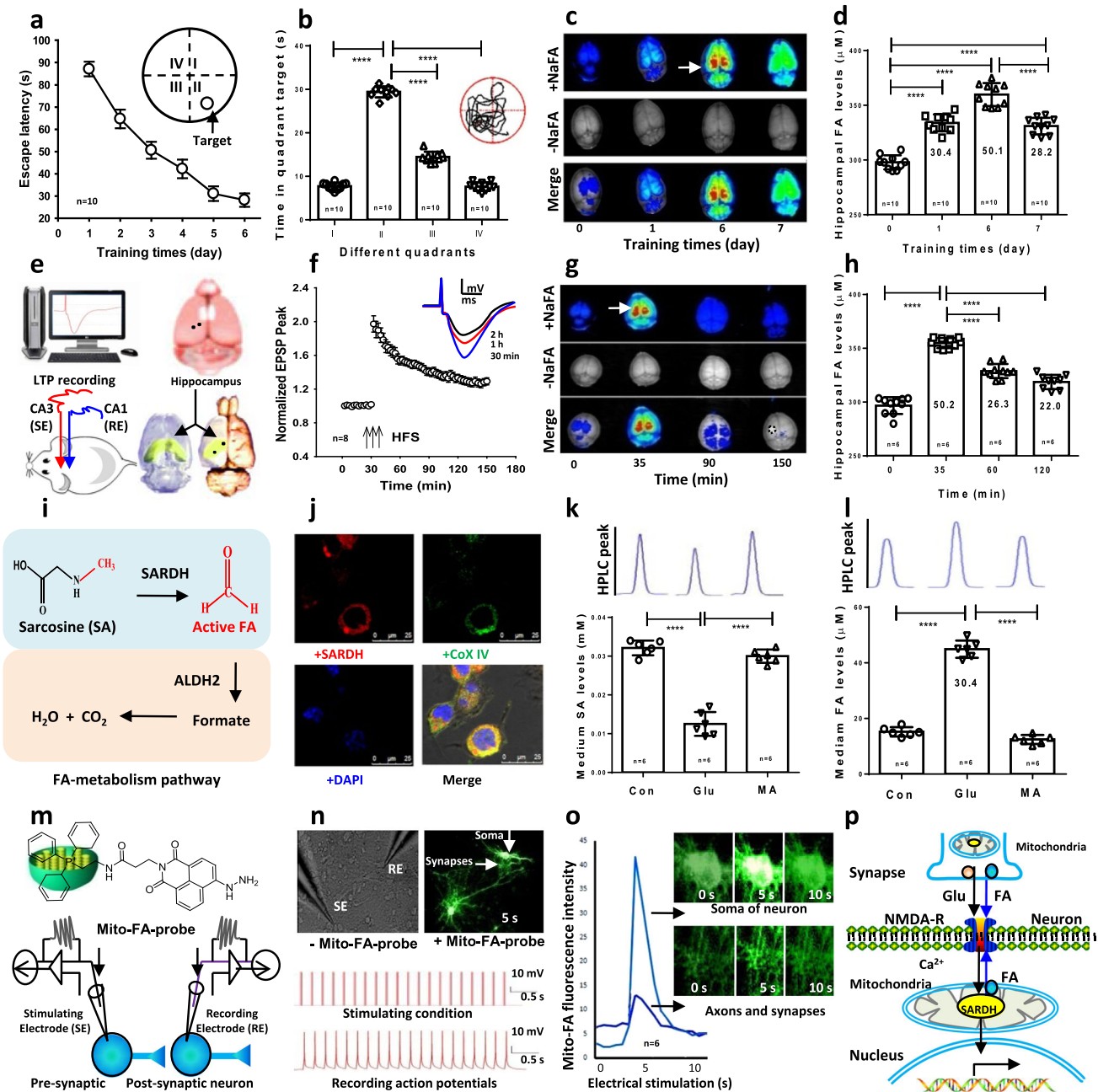

**Fig. 1** Spatial learning elicits formaldehyde generation. **a**, **b** Spatial learning and memory in wild-type SD rats trained in the MWM ($n = 10$ per group). **c** Brain formaldehyde fluorescence revealed by the in vivo imaging system. NaFA: a fluorescence probe of free formaldehyde, $n = 3$. **d** Hippocampal formaldehyde (FA) levels detected by Fluo-HPLC ($n = 10$). **e** An in vivo LTP recording in CA1 from Schaffer collateral stimulation and 3D views of the hippocampus (yellow). SE stimulating electrode, RE recording electrode. **f** Late-LTP (L-LTP) formation in vivo. $n = 8$, HFS high-frequency stimulation. **g** Brain formaldehyde revealed by the in vivo imaging system, $n = 3$. **h** Hippocampal formaldehyde levels detected by Fluo-HPLC ($n = 6$). **i** The pathway of formaldehyde metabolism. SARDH sarcosine dehydrogenase, ALDH2 aldehyde dehydrogenase. **j** Colocalization of SARDH (red) and the mitochondrial marker- Cox IV (green) in the cultured hippocampal neurons. DAPI: blue, a nuclear dye. **k**, **l** The cultured medium sarcosine and formaldehyde levels detected by Fluo-HPLC. SA sarcosine, MA methoxyacetic acid, an inhibitor of SARDH, $n = 6$. **m** Intracellular infusion of the mitochondrial formaldehyde probe (mito-FA-probe). **n** A train of 20 pulses in the presynaptic neurons induced multiple action potentials in the postsynaptic neurons, $n = 8$. **o** The active formaldehyde generated from mitochondria in the axon, synapse and soma of the cultured neurons imaged by the mito-FA probe, $n = 6$. **p** The model of endogenous formaldehyde-enhanced memory formation. The data are expressed as the mean ± standard error (s.e.m.). ***$p < 0.001$; ****$p < 0.0001$.

axons and somas of hippocampal neurons contain highly enriched mitochondria[24,25], and formaldehyde may modulate NMDA-R activity[9,10]. Hence, we speculated that the mitochondrial SARDH-derived formaldehyde regulates memory formation via NMDA-R (Fig. 1p).

**Formaldehyde precursors enhances LTP and memory formation.** To examine whether this elevated active formaldehyde has effects on synaptic function and/or memory, formaldehyde precursors and aqueous formaldehyde solutions were directly intracerebroventricular injection (i.c.v.) or intrahippocampally

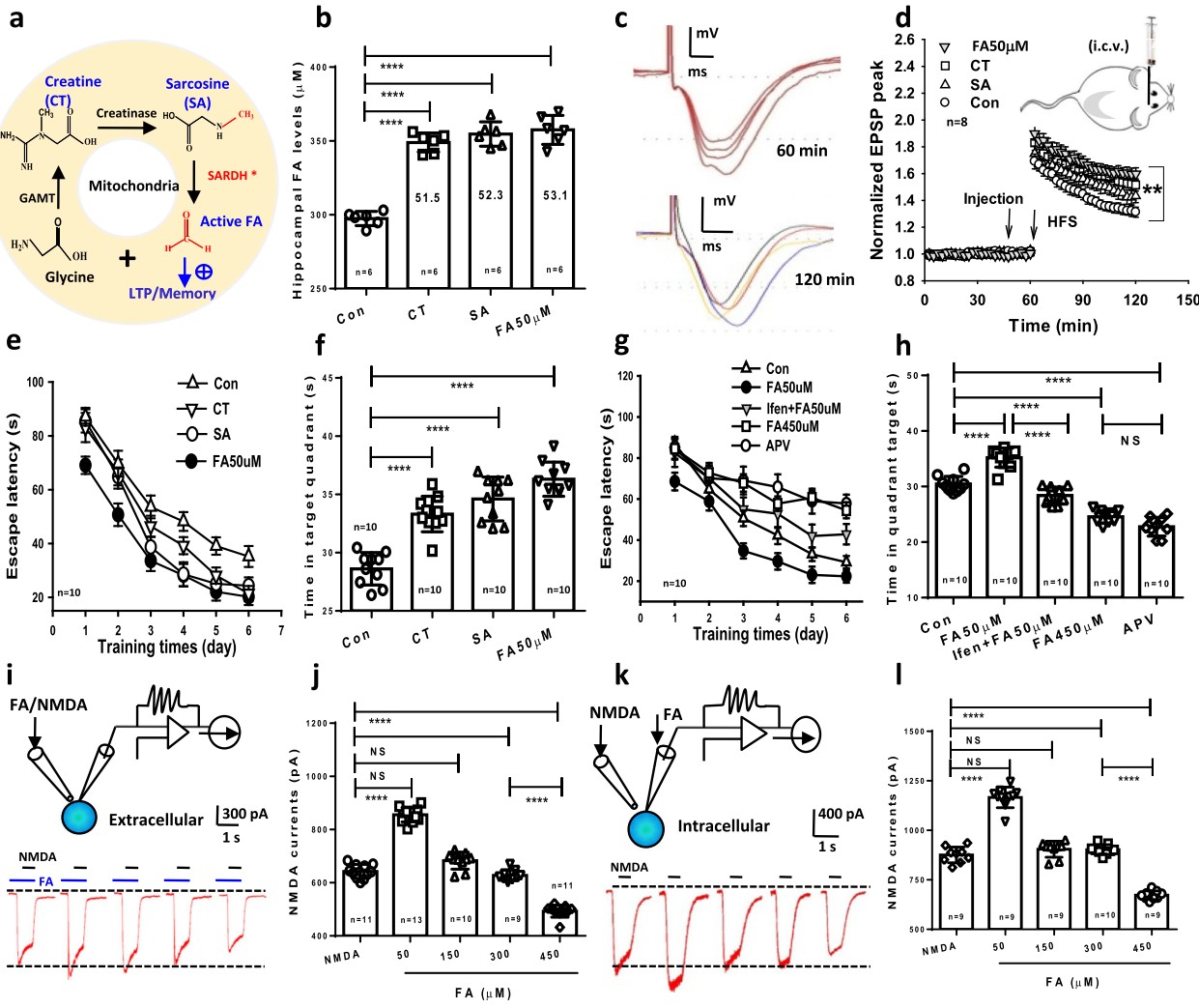

**Fig. 2 Formaldehyde regulates LTP and memory via NMDA-R. a** The metabolism pathway of formaldehyde precursors. GAMT guanidinoacetate methyltransferase, SARDH sarcosine dehydrogenase (a formaldehyde-generating enzyme), ALDH2 aldehyde dehydrogenase (a formaldehyde-degrading enzyme), FA formaldehyde, CT creatine, SA sarcosine. **b** Hippocampal formaldehyde levels quantified by HPLC-Fluo 30 min after i.c.v. injection with medicines—creatine (200 μM), sarcosine (200 μM), and formaldehyde (50 μM) ($n = 6$). **c, d** The fEPSP traces at 60 and 120 min (**c**) and enhancement of hippocampal long-term potentiation (LTP) (**d**) by infusion of above medicines ($n = 8$). **e** After 6 days of MWM training, repeated measures two-way ANOVA revealed a difference in group: $F_{(3, 36)} = 32.43$, $p < 0.001$, time: $F_{(5, 203)} = 74.98$, $p < 0.001$, and a group/time interaction: $F_{(15, 203)} = 6.32$, $p < 0.001$. Post hoc tests showed that the mean escape latency values for the CT- and SA-injected group were significantly shorter than the control group on days 4, 5 and 6, $t(36) = 5.17$, $p < 0.01$; and FA-injected group had shorter escape latency than the control group on days 1, 2, 3, 4, 5 and 6, $t(36) = 7.11$, $p < 0.01$, respectively. **f** The CT-, SA-, and FA-injected mice had longer staying times in target quadrant than control (wild-type mice) ($t(36) = 9.29$, $p < 0.001$, $n = 10$ mice per group). **g, h** The effects of Ifen and APV on spatial memory assayed by MWM ($n = 10$, rat per group). Ifenprodil (Ifen, 10 μM, a specific antagonist of NR2B); DL-2-amino-5-phosphonovaleric acid (APV, 50 μM, a nonspecific antagonist of NMDA-R). **i–l** The effects of extracellular or intracellular infusion of various formaldehyde concentrations on NMDA currents in the cultured hippocampal neurons ($n = 9 - 13$). Data are expressed as the mean ± standard error (s.e.m.). ***$p < 0.001$; ****$p < 0.0001$.

infused into the brains of healthy SD rats. Endogenous formaldehyde can be derived from foods, such as formaldehyde precursors: creatine and sarcosine. Creatine is transformed to sarcosine by creatinase[26], and mitochondrial SARDH enzymolyzes sarcosine into active formaldehyde[18,22] (Fig. 2a). After creatine, sarcosine, and formaldehyde (50 μM) were intrahippocampally microinjected 30 min, the hippocampi were taken out for detection of formaldehyde concentrations. We found that the elevated hippocampal formaldehyde levels were ~50 μM (Fig. 2b). To address whether these formaldehyde precursors enhance synaptic plasticity, we recorded hippocampal LTP in SD rats. The results showed that injection of creatine, sarcosine, and formaldehyde at 50 μM (i.c.v.[27]) before HFS 30 min all facilitated LTP formation in vivo (65 min-fEPSP peak for Con: 1.352 ± 0.037;

fEPSP peak for formaldehyde: 1.667 ± 0.0465; fEPSP peak for creatine: 1.634 ± 0.0508; fEPSP peak for sarcosine: 1.505 ± 0.045. $F_{(3, 69)} = 24.72$; $p < 0.001$) (Fig. 2c, d). Scavenging this elevated formaldehyde suppresses in vivo LTP formation[27]. These data demonstrate that endogenous formaldehyde is required for LTP induction.

We further observed the effects of the intrahippocampal infusion of formaldehyde precursors on spatial memory in rats in MWM. Acquisition of the location of the hidden platform, measured as the average latency to find the platform over several sessions of training, each separated by a day. The formaldehyde-, sarcosine-, and creatine-injected rats demonstrated significantly rapider acquisition compared with control (Fig. 2e). On day 7, the rats injected with creatine and sarcosine as well as formaldehyde

treatment had longer times in target quadrant than control rats ($n = 10$, $p < 0.01$) (Fig. 2f). Consequently, spatial learning activity-related formaldehyde generation ($\Delta$ ~50 μM) contributes to the enhancement of LTP and memory formation.

**Formaldehyde-regulated NMDA-R controls memory formation.** The amino acid glutamate (Glu) is recognized to be the main excitatory neurotransmitter in the brains. Glu exerts excitotoxic activity by mainly activating NMDA-type glutamate receptor (NMDAR). This receptor has been the most extensively studied and the most frequently implicated in central nervous system diseases[28]. NMDA-R activity is critical for hippocampus-dependent spatial memory[11,29], and formaldehyde has been shown to modulate NMDA-R activity[9,10]; thus we hypothesized that antagonists of NMDA-R could have an effect of formaldehyde-associated spatial memory. After 6 days of MWM training, repeated measures two-way ANOVA revealed a difference in group: ($F_{(4, 20)} = 37.09$, $p < 0.001$), training day ($F_{(5, 25)} = 12.18$, $p < 0.001$), and a group × time interaction ($F_{(20, 100)} = 6.45$, $p < 0.001$). Post hoc tests showed that the mean escape latency values for excess formaldehyde-injected rats (450 μM) were longer than control (wild-type rats) on day 3 ($F_{(4, 20)} = 5.12$, $p = 0.003$), day 4 ($F_{(4, 20)} = 6.24$, $p = 0.001$), day 5 ($F_{(4, 20)} = 7.15$, $p = 0.003$), and day 6 ($F_{(4, 20)} = 8.70$, $p = 0.001$); however, the rats injected with 50 μM had shorter escape latency than control group on days 1, 3, 4, 5 and 6, respectively ($p < 0.001$; $n = 10$, rat per group). Intrahippocampal injection of ifenprodil (Ifen, 10 μM, a specific antagonist of NR2B), statistically significant reduced 50 μM formaldehyde-enhanced spatial learning and memory in SD rats (Fig. 2g, h). Meanwhile, we also observed that intrahippocampally infusion of DL-2-amino-5-phosphonovaleric acid (APV, 50 μM, a nonspecific antagonist of NMDA-R) as well as 450 μM formaldehyde injection did impair learning and memory in SD rats ($p < 0.01$, Fig. 2g, h). These data suggest that exogenous formaldehyde most likely affects memory via NMDA-R.

We then investigated whether exogenous and aqueous formaldehyde directly modulates NMDA currents in the cultured hippocampal neurons. As expected, both intracellular and extracellular infusion of formaldehyde at the active level (50 μM) enhanced NMDAR-mediated currents, while 150−300 μM formaldehyde had no effect. Notably, a higher concentration (450 μM) of formaldehyde inhibited NMDA- currents (Fig. 2i–l). Hence, exogenous formaldehyde bidirectionally modulates NMDAR activity.

**Formaldehyde dually regulates NR1/NR2B via cysteine residues.** To address the possible molecular mechanisms that this elevated formaldehyde (~50 μM) regulates NMDA-R, we transfected GFP-NR1a and NR2B plasmids into Chinese hamster ovary cells (CHO) and tested the intracellular $Ca^{2+}$ influx ($[Ca^{2+}]i$) by laser scanning confocal microscopy (Supplementary Fig. 1a). The results showed that formaldehyde at 50 μM alone did not evoke $Ca^{2+}$ influx; however, it enhanced NMDA-evoked $[Ca^{2+}]i$ elevation. The specific NR2B antagonist ifenprodil (Ifen) could suppress this enhancement (Supplementary Fig. 1b), suggesting that NR2B may be the target of formaldehyde at 50 μM. Previous studies have shown that the tyrosine (Y) 231 and cysteine (C) 232 residues of NR2B are the specific binding sites for Ifen (3-dimensional (3D) crystal structure of NR1/NR2B complex, PBD ID: 3QEL)[30,31] (Fig. 3a, b and Supplementary Fig. 2a), and formaldehyde spontaneously have reaction with cysteine (C)[32] (Fig. 3c). We speculated that Ifen prevents formaldehyde-binding to C232, thereby blocking formaldehyde-dependent facilitation of NMDAR activity (Supplementary Fig. 2b, c). Therefore, deleting the ~400-amino acid of amino-

terminal domain (ATD) containing C232 (Supplementary Fig. 2d, e), or creating a single point mutation (C232A) in NR2B (the DNA sequences of the plasmid of NR2B with C232A mutation were identified by gene sequencing, Supplementary Fig. 3), was performed to identify that C232 residue in the ATD sequence is the target site for reaction with formaldehyde. Clearly, deleting ATD sequence of NR2B (D-ATD) reduced formaldehyde-induced enhancement of NMDA currents in the CHO cells transfected with plasmid of GFP-NR1/NR2B-D-ATD (Fig. 3d, e). This result suggests that the target residue of formaldehyde-activated NMDA-R may be at the ATD region. Further, we mutated the 232 Cysteine (C232) to Alanine (C232A) in the ATD structure, and found that formaldehyde-induced enhancement of NMDA currents was markedly reduced in the CHO cells transfected with plasmid of GFP-NR1/NR2B-C232A ($p < 0.01$) (Fig. 3d, e). These two critical evidences indicate that this C232 residue in ATD sequence of NR2B is the target site for formaldehyde (50 μM)-enhancing NMDA currents.

Injection of formaldehyde over 450 μM impairs spatial memory in healthy rats; we speculated that 450 μM formaldehyde may close or inactivate NMDA-R. A previous study found that there is cross-linking between residue C79 of NR1 and 80 lysine (K80) of NR2A[33]. Actually, the C79 residue of NR1 and K79 of NR2B is not directly connected in the 3D crystal structure of NR1/NR2B complex (PBD ID: 4PE5)[34]; however, the short distance (<10 Å) of these two residues is enough to be linked by some small molecules (Fig. 3f). Our above data showed that formaldehyde could affect NR1/NR2B activity, and formaldehyde at a sufficient concentration can act as a cross-linker between the residues of proteins: C-C, C-K and C-Y[35,36]. This is a reasonable speculation that formaldehyde at 450 μM may close or suppress NMDA-R by cross-linking C79 of NR1 to K79 of NR2B (Fig. 3g and Supplementary Fig. 4). In fact, a single point mutation of NR1 C79S, NR2B K79S, and the combination of NR1 C79S/NR2B K79S (the DNA sequences of these two plasmids were identified by gene sequencing, Supplementary Figs. 5 and 6) reversed formaldehyde-mediated inhibition of NMDA currents in the transfected CHO cells, respectively (Fig. 3h, i). These data confirm that excessive formaldehyde suppresses NMDA-R activity by cross-linking NR1 to NR2B residues.

**Formaldehyde overload impairs memory in $Aldh2^{-/-}$ mice.** Our above data indicate that exogenous formaldehyde dually regulates memory via NMDA-R. To address the critical question whether endogenous formaldehyde also affects memory, we deleted $Aldh2$ gene to artificially induce formaldehyde accumulation in the brains of $Aldh2^{-/-}$ mice (Fig. 4a) and assessed the spatial memory behaviors in MWM. Analyses of escape latency within each training day revealed that these 7-month-old $Aldh2^{-/-}$ mice had significantly longer escape latency than WT mice ($n = 10$, $p < 0.01$), indicating a marked impairment in the ability of spatial learning (Fig. 4b). Moreover, $Aldh2^{-/-}$ mice exhibited an obvious decline in spatial recall in the probe test, and neurochemical analyses confirmed elevated formaldehyde (approached maximum 500 μM) compared to WT mice (~300 μM) (Fig. 4c, d). In addition, these $Aldh2^{-/-}$ mice also showed a decrease in exploratory activity toward a novel object compared to WT mice in the open-field novel object recognition (NOR) test (Supplementary Fig. 7).

Then we investigated whether intragastric administration of 500 μM L-cysteine (L-cys, a formaldehyde scavenger[20,21]) reduces brain formaldehyde concentrations and rescues memory deficits in healthy adult wild-type rats. After 6 days of MWM training, repeated measures two-way ANOVA revealed a difference in group: ($F_{(2, 27)} = 11.36$, $p < 0.001$), training day ($F_{(5, 152)} = 43.78$, $p < 0.001$), and a group × time interaction ($F_{(10, 152)} = 8.54$,

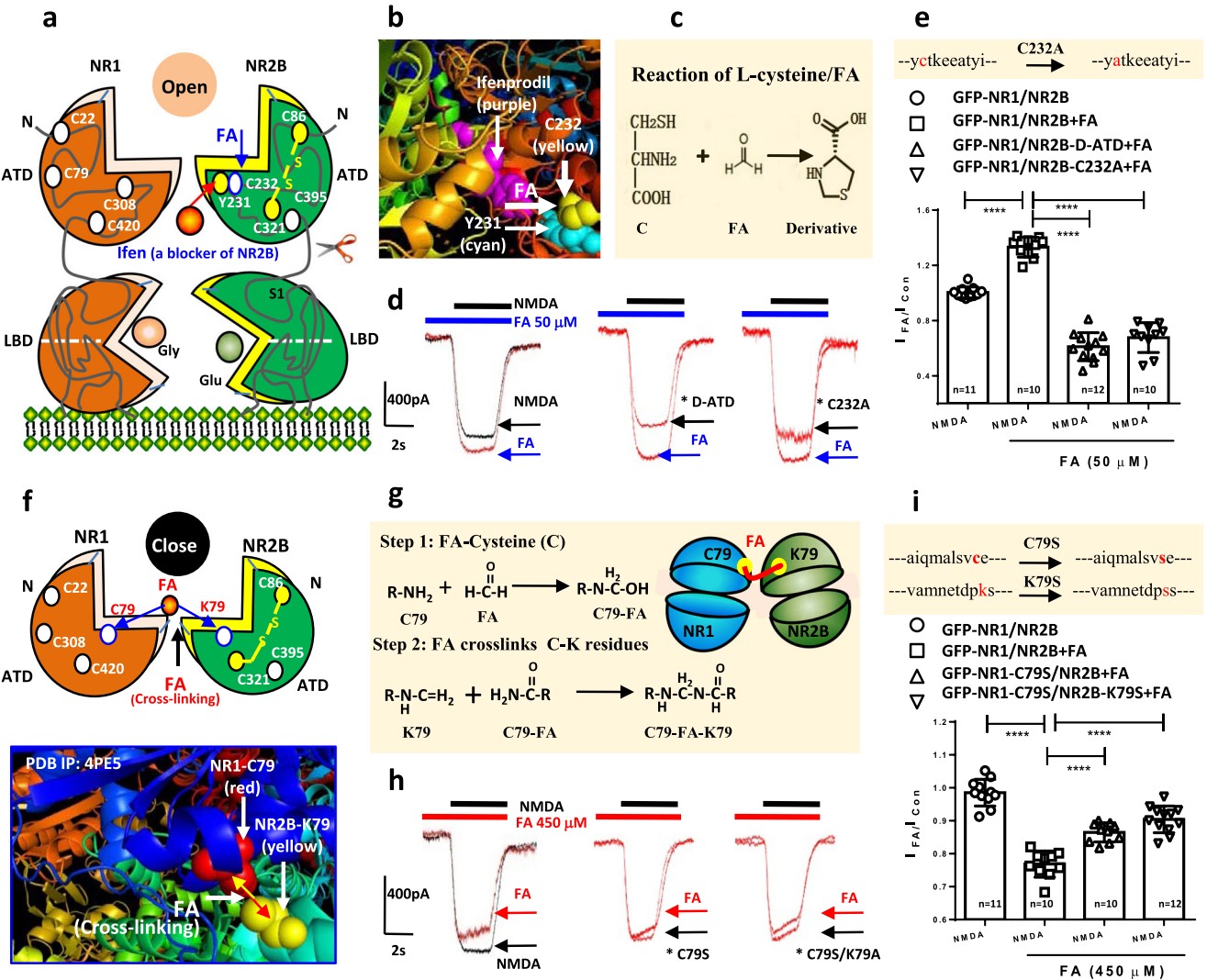

**Fig. 3 Formaldehyde dually regulates NMDA-R via different subunits. a, b** The model of active formaldehyde-activated NMDA-R by binding to C232 residue of NR2B. Ifenprodil (Ifen, a specific antagonist of NR2B). FA formaldehyde. **c** The chemical equation between formaldehyde and L-cysteine. **d, e** Recording of NMDA currents in CHO cells cotransfected with GFP-NR1a and NR2B without the amino terminal domain (NR2B-ATD) or a single point mutation (C232A). LBD ligand binding domain. $n = 10–12$. **f, g** The model of excessive formaldehyde-suppressed NMDA-R by cross-linking C79 of NR1 to K79 of NR2B. **h, i** Recording of NMDA currents in CHO cells cotransfected with C79S mutant NR1a and/or K79S mutant NR2B, respectively ($n = 10–12$). The data are expressed as the mean ± standard error (s.e.m.). ***$p < 0.001$; ****$p < 0.0001$.

$p < 0.001$). Post hoc tests showed that the mean escape latency values for formaldehyde-injected rats were a significant longer than control (wild-type rats) on day 2 ($F_{(2, 27)} = 2.15$, $p = 0.001$), day 3 ($F_{(2, 27)} = 3.22$, $p = 0.003$, day 4 ($F_{(2, 27)} = 6.25$, $p = 0.001$), day 5 ($F_{(2, 27)} = 5.72$, $p = 0.002$), and day 6 ($F_{(2, 27)} = 6.07$, $p = 0.001$); while the escape latency in these rats injected with formaldehyde and L-cys was shorter than formaldehyde-injected rats on days 4, 5 and 6 ($p < 0.01$; $n = 10$ rat per group) (Fig. 4e). Treatment of L-cys for consecutive 7 days reversed excess formaldehyde injection-induced formaldehyde overload and memory impairments in healthy SD rats (Fig. 4f, g). These data confirm that endogenous formaldehyde overload directly causes memory decline in rats.

**Formaldehyde is overloaded in AD patients.** Our previous study showed that urine formaldehyde levels were negatively correlated with cognitive function in AD patients[8]. To establish the relationship between formaldehyde metabolism and ALDH2 activity

(a formaldehyde-degrading enzyme) or cognitive decline, the cognitive abilities of 158 participants were examined using the Mini-Mental State Examination (MMSE[8]) together with analysis of the genotype of *ALDH2* and urinalysis of formaldehyde (Supplementary Table 1). Consistent with a formaldehyde overload causing cognitive impairment, urine formaldehyde levels were negatively correlated with MMSE scores (Fig. 4h). Further, the activity of ALDH2 was about fivefold lower in the blood of AD patients than age-matched healthy controls (Fig. 4i), and magnetic resonance imaging (MRI) revealed marked atrophy in the left prefrontal lobe and significant ventriculomegaly (marked white triangle) compared to healthy controls (Fig. 4j, k). Genotype analysis by PCR showed that these patients harbored three genotypes of the *ALDH2* G487A variant: normal or typical homozygote *ALDH2[1*1]* (GG), heterozygote *ALDH2[1*2]* (GA), and homozygous mutant *ALDH2[2*2]* (AA). Genotype distribution was consistent with Hardy–Weinberg equilibrium (Supplementary Table 2). In fact, h*ALDH2[2*2]* (AA) mutation has been found to induce ALDH2 inactivity[14]. Thus, combined with above results of

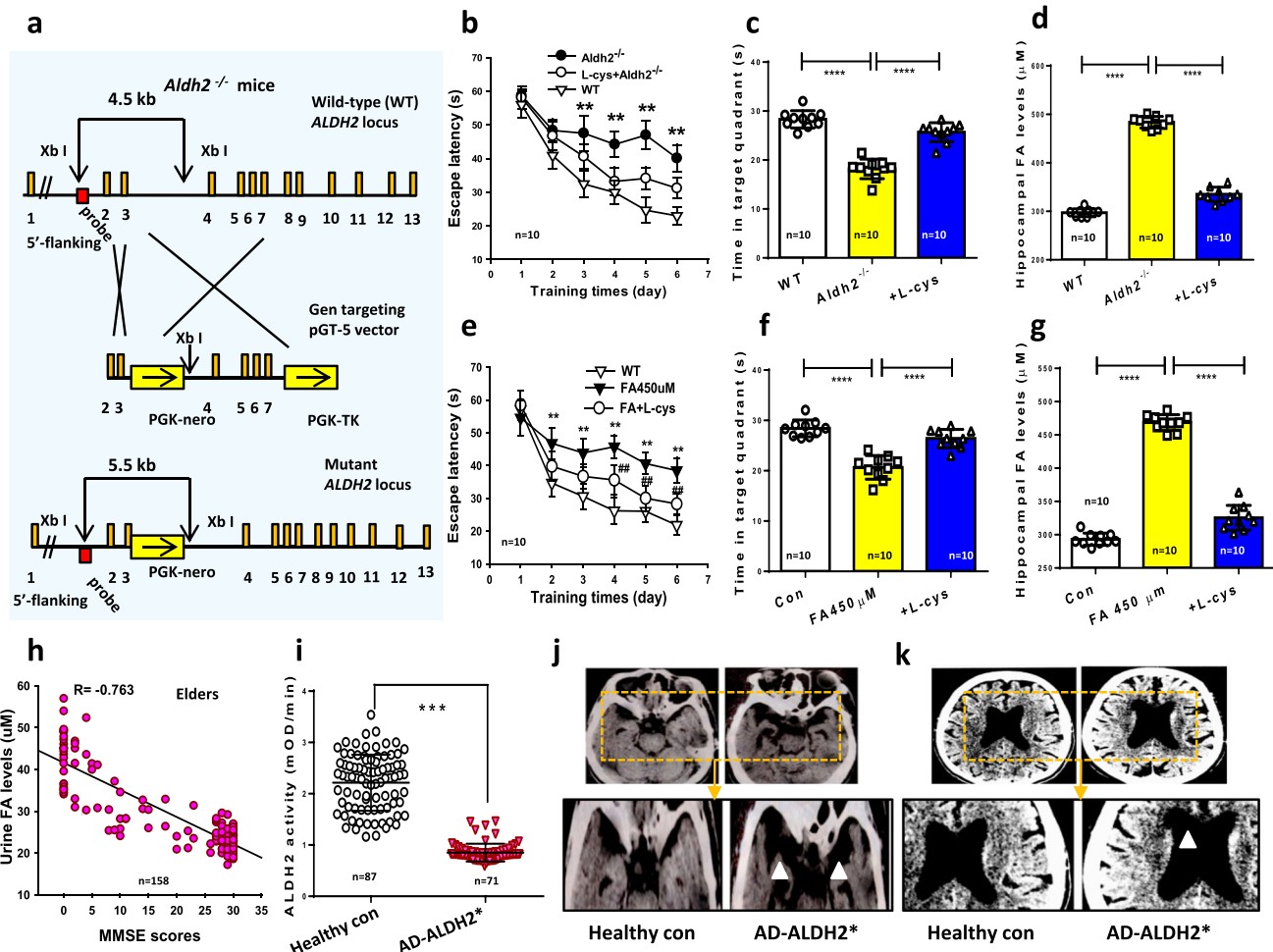

**Fig. 4** *ALDH2 mutation-induced formaldehyde overload causes amnesia.* **a** The scheme for generation of $Aldh2^{-/-}$ mice. **b** After 6 days of MWM training, repeated measures two-way ANOVA revealed a difference in group: ($F_{(2, 27)} = 14.19$, $p < 0.001$), training time ($F_{(5, 152)} = 58.32$, $p < 0.001$), and a group × time interaction ($F_{(10, 152)} = 4.91$, $p < 0.001$). Post hoc tests showed that the mean escape latency values for $ALDH2^{-/-}$ mice were a significant longer than control (wild-type mice) on day 3 ($F_{(2, 27)} = 3.86$, $p = 0.005$), day 4 ($F_{(2, 27)} = 4.61$, $p = 0.004$), day 5 ($F_{(2, 27)} = 7.13$, $p = 0.002$), and day 6 ($F_{(2, 27)} = 6.27$, $p = 0.001$); while the escape latency in $ALDH2^{-/-}$ mice with L-cys treatment was no statistically significant difference than control from days 1 to 5 ($p > 0.05$; $n = 10$ mice per group). **c** $ALDH2^{-/-}$ mice with L-cys injection reversed the reduced time in target quadrant of these mutated mice without formaldehyde treatment ($t (27) = 6.25$, $p < 0.001$). **d** Hippocampal formaldehyde (FA) concentrations detected by Fluo-HPLC ($n = 10$). **e, f** Spatial learning and memory assayed by the MWM in WT rats with or without intrahippocampal infusion of excessive (450 μM; $t (27) = 11.60$, $p = 0.002$) formaldehyde or formaldehyde scavenger-L-cysteine (L-cys: 500 μM) ($t (27) = 1.49$, $p = 0.165$); $n = 10$ mice/group. **g** Hippocampal formaldehyde levels detected by Fluo-HPLC ($n = 10$). **h** Negative relationship between urine formaldehyde and MMSE scores in 158 elderly AD patients. **i** ALDH2 activity analyzed by a human ALDH2 kit ($p < 0.01$). **j** Prefrontal lobe atrophy revealed by MRI. **k** Ventriculomegaly in AD patients with *ALDH2* mutation. The data are expressed as the mean ± standard error (s.e.m.). ***$p < 0.001$; ****$p < 0.0001$.

$Aldh2^{-/-}$ mice, hALDH2 with low-activity variants or loss-of-function mutations lead to formaldehyde overload and dementia in AD patients.

We also observed the effects of formaldehyde at different concentrations on cell viability in the cultured human SY5Y cells. The results showed that formaldehyde at 50 μM enhanced cell viability, while formaldehyde at 150 and 300 μM had no effect. However, 6-h formaldehyde incubation at 450 μM clearly reduced cell viability (Supplementary Fig. 8). This result indicates that excess formaldehyde induces neuron death which contributes to cortical atrophy.

**Formaldehyde is deficient in sarcosinemia children**. Endogenous formaldehyde overload caused dementia in patients; we further want to know whether endogenous formaldehyde deficiency also affects human cognition. To answer this question, we examined the relationship between formaldehyde deficiency and

intelligence quotient in children with sarcosinemia, a rare pediatric disease associated with the mutation of *SARDH* (a formaldehyde-generating enzyme[18]) (Supplementary Table 3). Intelligence quotient was assessed in 11 pediatric sarcosinemia patients using the Wechsler Intelligence Scale for Children (WISC)[37] together with urinalysis for formaldehyde and blood SARDH activity measures. The results revealed a positive correlation between cognitive ability and urine formaldehyde in these children (Fig. 5a). Further, sarcosinemia patients exhibited lower SARDH activity in blood and lower formaldehyde concentrations in urine than age-matched healthy controls (Fig. 5b, c). High-throughput gene sequencing identified SARDH mutations at c.1553, c.1540 and c.860 (Supplementary Fig. 9). The results of magnetic resonance imaging (MRI) and Fluorine[18]-fluorodeoxyglucose positron emission tomography/computed tomography ($^{18}$F-FDG PET/CT) revealed substantial hippocampal atrophy (Fig. 5d, e). This finding was consistent with previous reports that

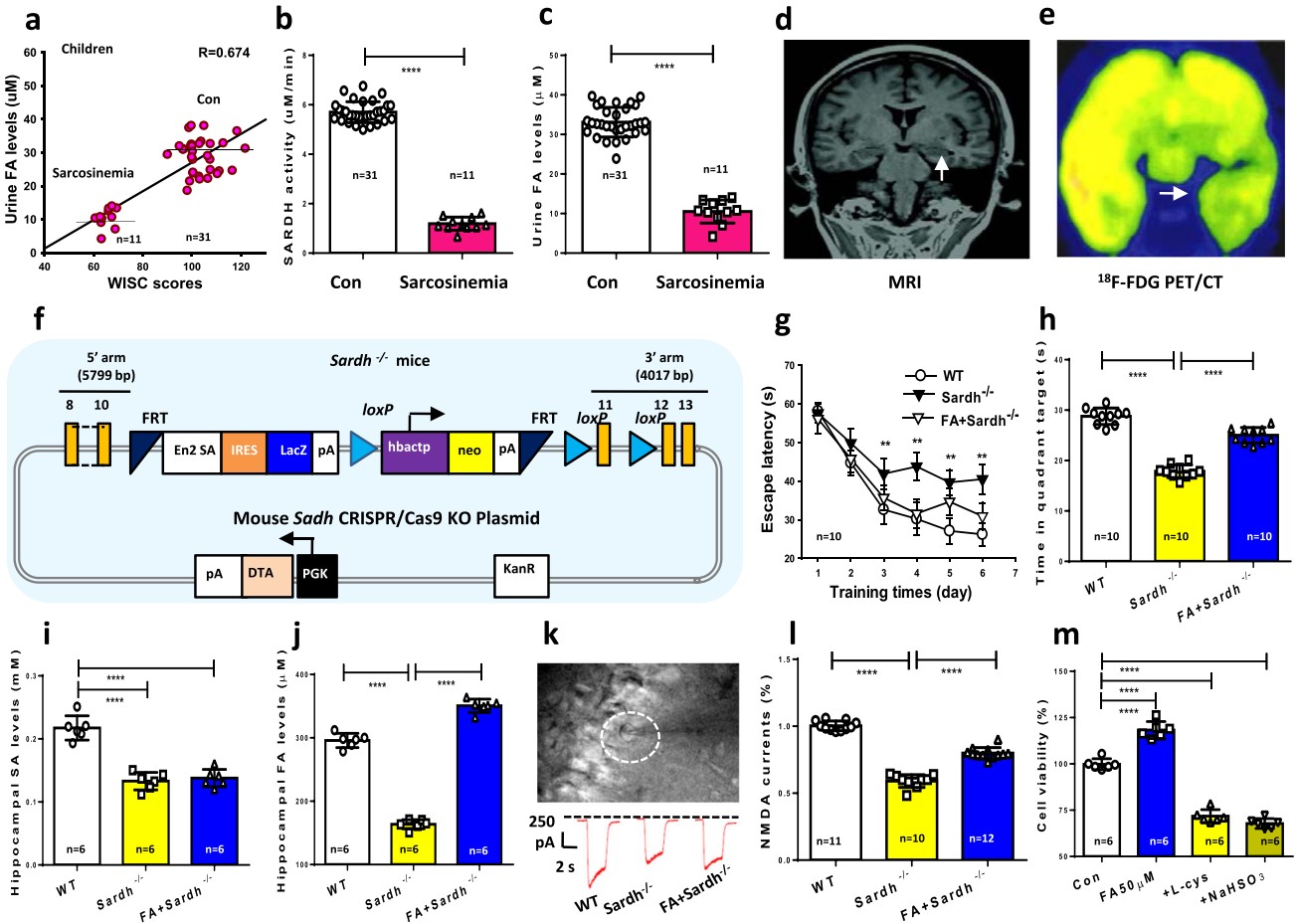

**Fig. 5** *SARDH* mutation-induced formaldehyde deficiency causes amnesia. **a** Positive relationship between urine formaldehyde (FA) levels and Wechsler Intelligence Scale for Children (WISC) scores in 11 pediatric sarcosinemia patients and 31 healthy children. **b** Blood SARDH activity analyzed by a human SARDH kit ($p < 0.01$). **c** Urine formaldehyde concentrations detected by Fluo-HPLC ($p < 0.01$). **d** Right hippocampal atrophy revealed by MRI. **e** Impaired glucose metabolism in the right hippocampus revealed by [18]F-FDG PET/CT. **f** The scheme for generation of *Sardh*$^{-/-}$ mice. **g** After 6 days of MWM training, repeated measures two-way ANOVA revealed a difference in group: ($F_{(2, 27)} = 16.38$, $p < 0.001$), training time ($F_{(5, 152)} = 72.54$, $p < 0.001$), and a group × time interaction ($F_{(10, 152)} = 2.45$, $p < 0.001$). Post hoc tests showed that the mean escape latency values for *SARDH*$^{-/-}$ mice were a significant longer than control (wild-type mice) on day 3 ($F_{(2, 27)} = 2.41$, $p = 0.001$), day 4 ($F_{(2, 27)} = 4.57$, $p = 0.005$), day 5 ($F_{(2, 27)} = 6.39$, $p = 0.003$), and day 6 ($F_{(2, 27)} = 6.43$, $p = 0.002$); while there was no statistically significant difference in escape latency between *SARDH*$^{-/-}$ mice with 200 μM formaldehyde injection and control ($p > 0.05$; $n = 10$ mice per group). **h** *SARDH*$^{-/-}$ mice with formaldehyde infusion reversed the reduced time in target quadrant of these mice without formaldehyde treatment ($p < 0.001$). **i** Hippocampal sarcosine levels detected by a mouse sarcosine kit, $n = 6$. **j** Hippocampal formaldehyde levels detected by Fluo-HPLC, $n = 6$. **k**, **l** NMDA currents recording in the brain slices of *Sardh*$^{-/-}$ mice and formaldehyde-injected *Sardh*$^{-/-}$ mice, $n = 10–12$. **m** The effects of formaldehyde scavengers- L-cys (300 μM) and NaHSO$_3$ (300 μM) on the cell viability of the cultured hippocampal neurons detected by CCK-8 kit, $n = 6$. Data are expressed as the mean ± standard error (s.e.m.). ***$p < 0.001$; ****$p < 0.0001$.

hippocampus impairments lead to memory deficits in children[38]. Thus, *SARDH* mutation-induced formaldehyde deficiency may be related with human cognitive impairments.

**Formaldehyde deficiency induces amnesia in *Sardh*$^{-/-}$ mice.** To further provide evidence that *SARDH* mutation directly impairs human cognition, we artificially made endogenous formaldehyde deficiency by knockout of *Sardh* gene in mice (Fig. 5f). The *Sardh*$^{-/-}$ mice exhibited body weight decline and low expression of SARDH in the hippocampus than control mice and were associated with no difference in motor ability than control (Supplementary Fig. 10). MWM tests also revealed deficits in spatial learning in *Sardh*$^{-/-}$ mice compared to wild-types (WTs) and mutants receiving exogenous formaldehyde. After the 6-day MWM training, analyses of escape latency within trails for each day revealed that *Sardh*$^{-/-}$ mice exhibited longer escape latencies than WT mice ($p < 0.01$); however, *Sardh*$^{-/-}$ mice receiving

formaldehyde injection exhibited a shorter escape latency than the mutated mice without formaldehyde treatment (Fig. 5g, h). Biochemical assays confirmed that *Sardh* knockout led to brain sarcosine and formaldehyde deficiency, while formaldehyde application increased brain formaldehyde from ~150 to ~354 μM (Fig. 5i, j). Consistent with memory decline, NMDA currents in the brain slices of *Sardh*$^{-/-}$ mice were markedly lower than control mice (Fig. 5k, l), indicating that endogenous formaldehyde deficiency causes memory deficits by suppressing NMDA currents.

We also observed the effects of formaldehyde scavengers on cell viability in the cultured hippocampal neurons at 24 h. The results showed that formaldehyde scavengers, L-cys (300 μM) and NaHSO$_3$ (300 μM), markedly reduced cell viability (Fig. 5m). This result indicates that chemically scavenging formaldehyde induces formaldehyde deficiency and neuron death, which may contribute to hippocampal atrophy.

Next, we used another chemical method to artificially reduce brain formaldehyde concentrations by intrahippocampally infusing $NaHSO_3$ (a formaldehyde scavenger[19]) in healthy SD rats and again examined spatial learning in the MWM. Analyses of escape latency within each day revealed a significant effect on treatment day 4 ($F_{(1, 18)} = 3.68$, $p = 0.003$), day 5 ($F_{(1, 18)} = 4.73$, $p = 0.001$), and day 6 ($F_{(1, 18)} = 5.02$, $p = 0.005$). Post hoc analyses for days 4, 5 and 6 showed that formaldehyde scavenger-injected mice exhibited significantly longer escape latencies than WT mice ($n = 8$, $p < 0.01$) (Supplementary Fig. 11a). On day 7, formaldehyde scavenger-injected mice exhibited an obvious decline in both brain formaldehyde and the ability of spatial recall in the probe trial compared to untreated rats (Supplementary Fig. 11b, c). Taken together, both *Sardh* deletion and chemically scavenging formaldehyde indeed induce formaldehyde deficiency-related amnesia in mice.

## Discussion

In this study, we found that spatial learning, HFS and glutamate stimulation elicited a rapid generation of active formaldehyde in rat hippocampus; the concentrations attained were sufficient to facilitate NMDA currents and enhanced LTP and memory formation. However, excess formaldehyde impaired memory in $aldh2^{-/-}$ mice and AD patients with *ALDH2* mutation by suppressing NMDA-R. In addition, brain formaldehyde deficiency in sarcosinemia children associated with *SARDH* mutation or in $Sardh^{-/-}$ mice also led to cognitive deficits by reducing NMDA currents (Supplementary Fig. 12). These findings suggest that endogenous formaldehyde is a promising candidate memory-related molecule.

A previous study reported that urine formaldehyde levels were negatively correlated with cognitive function in AD patients[8]. Overexpression of semicarbazide-sensitive amine oxidase (SSAO, a serum formaldehyde-generating enzyme) and abnormally high formaldehyde has been found in APP/PS1 mice and autopsy hippocampus from AD patients[39,40]. Degradation of formaldehyde is mainly dependent on mitochondrial ALDH2 and cytoplasmic alcohol dehydrogenase (ADH3, also named FDH)[23]. FDH is specifically expressed in the hippocampus and cortex[41]. Knockout of *FDH* leads to visual memory deficits in *Drosophila*[42]. Under glutathione depletion in AD[43], ALDH2 ($K_m$ value 500 μM for formaldehyde) plays a more important role in degrading pathological levels of formaldehyde[44]. Human *ALDH2* mutation also increases the risk for late-onset AD[45] and induces cognitive decline in diabetic patients[14]. In the current study, cognitive deficits were observed in formaldehyde-injected mice, $aldh2^{-/-}$ mice, and AD patients with *ALDH2* mutation. Excessive formaldehyde impaired NMDA-R signaling, thereby reducing the synaptoplastic changes underlying learning. Notably, memantine (an antagonist of NR2B) could alleviate dementia at the early stage of AD, which may prevent formaldehyde-inactivated NMDA-R. However, high formaldehyde concentrations caused neuron death, contributing to the cortical atrophy observed in AD. Use of a formaldehyde scavenger may be a potential therapeutic option for AD.

Formaldehyde deficiency and low intelligence quotient were observed in 11 pediatric sarcosinemia patients. These preschool children received insufficient clinical attention until they were tested by intelligence quotient screening tools and gene sequencing, mainly because sarcosinemia is a rare pediatric disease that is difficult to diagnose. Under physiological conditions, mitochondrial SARDH is an active formaldehyde-generating enzyme[18]. However, the children harboring a *SARDH* mutation exhibited a decline in SARDH activity and formaldehyde generation. Injection of a formaldehyde scavenger or *Sardh* knockout

in rodents induced formaldehyde deficiency as well as memory impairments. Endogenous formaldehyde also participates in one-carbon metabolism[46] and DNA methylation[6]. Dynamic changes in brain DNA methylation are necessary for new memory formation[47]. Hence, formaldehyde deficiency could induce neuronal metabolic disruption, hippocampal atrophy, and ensuing cognitive dysfunction.

Application of formaldehyde precursor creatine has been found to improve memory in rats[48]. Especially, oral supplementation of creatine significantly enhances human working memory[49]. Another precursor, sarcosine has been believed to enhance LTP formation in vitro[50]. In this study, we found that supplementation of formaldehyde precursors, creatine and sarcosine, could facilitate the formation of both LTP and memory, respectively. Application of L-cys (a formaldehyde scavenger) could reverse excessive formaldehyde-induced memory deficits. Interestingly, supplementation of a moderate amount of formaldehyde rescued formaldehyde deficiency-induced cognitive decline. Therefore, targeting formaldehyde and its metabolizing enzymes may be a promising therapeutic approach for preventing or treating cognitive disorders. In conclusion, our findings suggest that the primary small molecule, endogenous formaldehyde, regulates memory formation.

## Methods

**Clinical survey of sarcosinemia and dementia patients.** The clinical study was registered at the Chinese Clinical Trial Registry (http://www.chictr.org/cn, Unique Identifier: ChiCTR-OOC-14005576) and was conducted between October 2013 and July 2015. We recruited participants from Beijing geriatric hospital and Peking University Third Hospital, China. Participants who refused to provide blood or urine samples, had a life-threatening illness, or were unable to participate in the assessment were excluded from the entire survey. The mean age of the 158 individuals (71 elderly patients with Alzheimer's disease, AD and 87 age-matched controls) was $72.61 ± 3.19$ years, and cognitive function was assessed using the Mini-Mental State Examination (MMSE)[8] (Supplementary Table 1). The brains were scanned by *computer* tomography (CT). The mean age of the 11 preschool children with sarcosinemia and 31 age-match controls from Beijing Children's Hospital, Beijing BoAi Hospital, and Sichuan Provincial Hospital for Women and Children was $6.18 ± 1.69$ years, and human intelligence quotient was examined by using the Wechsler Intelligence Scale for Children (WISC)[51] (Supplementary Table 3). The brains were examined by the magnetic resonance imaging (MRI) and Fluorine[18]- fluorodeoxyglucose positron emission tomography/computed tomography ($^{18}$F-FDG PET/CT). Morning urine and serum from above participants were collected for measuring sarcosine and formaldehyde concentrations. The venous blood was used to enzyme activity analysis and gene sequencing. Informed consent was obtained from each participant either directly or from his or her guardian before participation. Ethical approval for the clinical investigations was obtained from the Clinical Ethics Committee at the Capital Medical University, China.

**Generation of $Sardh^{-/-}$ mice.** All protocols involving the use of animals were conducted in accordance with the Biological Research Ethics Committee, Capital Medical University, China. The adult male C56BL/6 mice at two-month-old age were housed in a temperature-controlled room under a 12-h light–12-h dark circadian cycle with access to water and food ad libitum. Mouse sarcosine dehydrogenase (SARDH: Gene ID: 192166) CRISPR/Cas9 KO plasmid (sc-407703) was purchased from Santa Cruz Biotechnology, USA. Further, $Sardh^{-/-}$ mice were made by the Institute of Medical Experimental Animals, Chinese Academy of Medical Sciences as shown in Fig. 5f. $Sardh^{-/-}$ mice were identified by using fluorescent histochemistry of SARDH (antibody of SARDH, SAB130499, 1:500, Sigma-Aldrich, USA), and by PCR with the DNA primers, F1: 5′-TGGCTCCA-CAAAACTCATGTC-3′; R1: 5′-GAGCCTCTGGATGGCTGAAG′, to generate a 84 bp product (Supplementary Fig. 10). The SARDH activity in mice was analyzed by mouse SARDH kit.

**Generation of $Aldh2^{-/-}$ mice.** $Aldh2^{-/-}$ mice were generated by the Beijing Biocytogen Co., Ltd, China and described in Fig. 4a. The *Aldh2* gene in the mice was identified by PCR with the DNA primers, F1: 5′-GGAAGATGTGGA-CAAGGCAG-3′; R1: 5′-ATGCCACTTTGTCCA CATCC-3′, to generate a 560 bp product. The ALDH2 activity was analyzed by mouse ALDH2 kit.

**Chemicals and reagents.** Unless otherwise stated, all chemicals and reagents were obtained from Sigma-Aldrich.

**Rotarod test**. The accelerating rotarod apparatus (AccuScan Instruments, Columbus, OH, USA) consisted of a suspended rod that accelerates at a constant rate, from 4 to 70 rpm in 300 s. At each day of training, mice were trained on the rotarod throughout a session of ten trials. A trial ends when the mouse falls off the rod or after reaching 300 s. Time was recorded for each trial. A resting time of 180 s was allowed between each trial.

**Intrahippocampal microinjection and Morris water maze test**. Stereotaxic surgery and intrahippocampal microinjection were conducted as described in previous report[52]. Briefly, the male adult Sprague−Dawley (SD) rats at 2-month-old age were implanted bilaterally with stainless steel guide cannula (−3.80 mm posterior to bregma; 2.2 mm left and right to the sagittal suture; −3.2 mm ventral to the skull surface) in the CA1 area of the hippocampus. Stainless steel styles (23-gauge) were inserted into the guide cannulae to keep them free of debris. After 7 days' recovery, these SD rats were randomly divided into different groups ($n = 6$ −10, each group): Control group (normal saline); formaldehyde groups (450 μM); formaldehyde scavenger-NaHSO₃ group (200 μM[53]). All injections were 5 μL in volume and slowly applied (5 min) and performed 30 min before Morris maze test. The behaviors of spatial memory in transgenic mice or SD rats were assessed by MWM as described previously[54,55].

**Open-field novel object recognition (NOR) task**. The NOR task consisted of 3 consecutive days of testing per trial; habituation, training with two identical objects, and testing with one familiar and one novel object. On the testing day, animals were allowed to explore the objects until they accumulated 30 s of total object exploration time (exploration was recorded when the nose of the mouse was within approximately 1 cm of the object). This was done rather than using a set exposure time in the test environment in order to account for any variability in movement and exploration that may occur between mice. Two measures of behavior were assessed: frequency of visits to the objects, and the time spent exploring each object. From the latter measure the discrimination index (difference in time exploring the novel and familiar object, divided by total exploration time) and the ratio of time spent with the novel object in relation to the familiar object were calculated.

**Intracerebroventricular injection and LTP recording**. Intracerebroventricular injection (i.c.v.) performed 30 min before high-frequency stimulation (HFS), hippocampal long-term potentiation (LTP) or late long-term potentiation (L-LTP) recordings from the Cornu Ammonis 1 (CA1) to Schaffer collateral pathway of CA3 in SD rats were performed as previously reported[56].

**Hippocampal neuron culture**. We prepared primary hippocampal neuron culture from 18-day-old embryos as described previously[57]. Then the medicines were coincubated or transfected into the cultured neurons. The reagents including: Methoxyacetic acid (MA, 200 μM, #08485, Sigma-Aldrich, USA), a competitive inhibitor of SARDH, which can inhibit sarcosine degradation[58].

**In vitro multiple action potentials recording**. Dissociated cell cultures of hippocampal neurons from 1- to 2-day-old Sprague−Dawley rats were prepared, and the bath and electrode solutions were used as previously described[59,60], except that the bath solution was not oxygenated. Recordings of multiple action potentials were done with two patch-clamp amplifiers (2 Axopatch 200B or 2 Axopatch 1D; Axon, Foster City, California). Signals were filtered at 5 kHZ using amplifier circuitry and stored on a VCR. Data were acquired using Axoscope 1.0 or pClamp 6.0 and analyzed using pClamp 6.0 software (Axon). Recordings were only made on neurons with an initial resting membrane potential ≤−50 mV. The multiple action potentials were recorded as described previously[61].

**Hippocampal slice preparation**. Hippocampal slices were prepared as described previously[62]. Following diethyl ether anesthesia, animals were decapitated, the brain was removed and cooled in artificial cerebrospinal fluid (ACSF, composition in mM: NaCl 134, KCl 5, NaH₂PO₄·2H₂O 1.5, MgSO₄ 2, CaCl₂ 2, NaHCO₃ 25, Glucose 10, saturated with 95% O₂/5% CO₂ with pH = 7.4) for 2 min. Then, the hippocampi were sliced to a thickness of 350−400 μm, using a vibrating tissue slicer (Vibratome 3000, Virratome, America). Slices were placed in a storage chamber filled with ACSF and kept at least for 1 h at room temperature (~25 °C). Before recording, the slices were individually transferred to a recording chamber and continuously perfused with oxygenated standard external solution at a rate of 2 mL/min during experiment, which contained (in mmol/L): 130 NaCl, 5.4 KCl, 1 MgCl₂, 2 CaCl₂, 10 HEPES, 10 glucose, and the pH was adjusted to 7.4 with Tris-base. Both external and internal solutions were made up using salts of analytical grade in twice-distilled deionized water.

**In vitro recording NMDA currents in hippocampal slices**. Whole-cell recordings were obtained, using an MultiClamp *700B Amplifier* (Molecular Devices, Union City, CA) as described previously[63]. Slices were superfused with the above external solution. Experiments were performed at ~25 °C, with the temperature maintained by an in-line heating device (Warner Instruments, Hamden, USA). Patch pipettes (2.0–3.5 MΩ) were filled with an internal solution containing the following (in mM):

Cs⁺ methanesulfonate 135, NaCl 8, HEPES 10, Cs-BAPTA 10, Mg-ATP 4, Na-GTP 0.4, verapamil 0.2 (voltage clamp experiments). NMDAR currents were recorded in the presence of D-serine (10 μM), NBQX (10 μM), TTX (0.5 μM), and picrotoxin (100 μM). The altered protocol was only that the holding voltage (+40 mV) was changed to hold at −60 mV. Flow-pipe solutions were continuously bubbled with 95% O₂, 5% CO₂.

**Site-directed mutagenesis of NR1 and NR2B plasmids**. The plasmids of pcDNA3.1-GFP-NR1a and pcDNA3.1-NR2B for NMDA-induced intracellular Ca²⁺ influx in the cultured CHO cells were donated from the lab of Dr. Jianhong Luo, Zhejiang University, China[64]. The wild-type plasmids of Grin1 and Grin2b were purchased from OriGene Technologies, Rockville, Maryland, USA. Deletion of sequences of ATD of NR2B, the single-site mutation of C79S of NR1, K79A of NR2B, or C232A of NR2B, according to the method of previous reports[30,65], were carried out by Beijing OriGene Technology Co., Ltd, Beijing, China. The DNA sequences of wild-type and mutants of NR1/NR2B subunit were identified by ABI 3730XL DNA analyzer (#3730XL, Thermo Fisher Scientific, USA).

**Transfection protocol**. Chinese hamster ovary cells (CHO) were grown in Ham's F-12 nutrient medium with 10% fetal bovine serum and 1 mM glutamine and passaged at a 1:10 dilution when 80% confluent, approximately every 2 days. CHO cells were seeded at $3 \times 10^5$ cells per well in six-well plates approximately 24 h before transfection with 1.3 mg of total DNA and 6 mL of Lipofectamine (GIBCO/BRL) in 1 mL of serum-free CHO media per 35-mm dish. The ratio of plasmids to total DNA was 1:4, and the ratio of NR1a to NR2B was 1:3. After a 4- to 5-h incubation at 37 °C, cells were refed with CHO medium containing 1 mM 5,7-dichlorokynurenic acid to prevent cell death. Cells were used for intracellular Ca²⁺ imaging or recording 40–50 h after transfection.

**Intracellular Ca²⁺ imaged by laser confocal**. The [Ca²⁺i] imaging in the CHO cells was used with Fluo-3 of Ca²⁺ probe. Another Ca²⁺ probe- Fluo-4 was used to the [Ca²⁺i] imaging in the cultured hippocampal neurons described as previously[66].

**NMDA currents recordings by patch-clamp**. Recordings were performed at ~25 °C on outside-out patches with 10−15 MΩ electrodes in the transfected CHO cells or hippocampal neurons. The extracellular solution contained 150 mM NaCl, 2.8 mM KCl, 1.0 mM CaCl₂, 10 mM Hepes, 0.01 mM glycine, pH adjusted to 7.2 with 0.3 N NaOH. The pipette solution contained 140 mM CsF, 10 mM EGTA, 1.0 mM CaCl₂, 10 mM Hepes, pH adjusted to 7.2 with CsOH. Recordings were made at −60 mV and 200 μM NMDA, 100 μM Ifenprodil (an antagonist of NR2B), and formaldehyde (0, 50, 150, 300 and 450 μM) were applied to the patch by complete bath.

**Detection of formaldehyde with formaldehyde kit or Fluo-HPLC**. Formaldehyde in the medium from cultured hippocampal neurons was incubated with different medicines was quantified by high-performance liquid chromatography with fluorescence detection (Fluo-HPLC) as described previously[20]. Formaldehyde concentrations in other samples were analyzed by DFOR-100 formaldehyde kit in accordance with the manufacturer's instructions (BioAssay Systems, Hayward, CA, USA).

**Detection of sarcosine with rat sarcosine kit or HPLC**. The sarcosine in the medium from cultures was collected and detected by Fluo-HPLC[67]. Sarcosine concentrations in other samples were analyzed by sarcosine Fluorimetric Assay Kit according to the manufacturer's instructions (ABIN411704, 4A Biotech Co., Ltd, China).

**Molecular simulations of formaldehyde-binding with NR1/NR2B**. The three-dimensional (3D) crystal structure of Glun1 and Glun2b in Complex with Ifenprodil (PDB ID: 3QEL) was downloaded for free (Protein Data Bank, https://www.rcsb.org/structure/3QEL). Crystal structure of Glun1a/Glun2b of Nmda Receptor Ion Channel (PDB ID: 4PE5) was downloaded (Protein Data Bank, https://www.rcsb.org/structure/4PE5). The possible binding site between formaldehyde and NR1/NR2B was analyzed by using the PyMOL 1.7 software, which can be downloaded for free (http://sourceforge.net/projects/pymol/).

**Statistics and reproducibility**. All data were tested for normality by the Kolmogorov−Smirnov test. When data were normally distributed, the statistical significance of differences was assessed with the unpaired $t$ test and one- or two-way ANOVA, analyzed by Tukey post hoc. When data were not normally distributed, the statistical significance of differences was judged on the basis of $p$ values with the Mann–Whitney $U$ test and analyzed by IBM SPSS v19.0 (SPSS Inc., Chicago, IL, USA).

Human serum biochemical index was assessed using Student's unpaired $t$ test. Gender of participants was assessed using the chi-squared test.

Correlations between urine formaldehyde levels and MMSE or WISC scores were assessed using Pearson correlation coefficient, both without adjustment, then accounting for sex and age. These data were analyzed by IBM SPSS v19.0.

Mice serum biochemical index was assessed using Student's unpaired $t$ test. The spatial memory behaviors in MWM of mice were analyzed by Tukey post hoc with repeated measures ANOVA.

The changes in the response amplitudes of LTP were analyzed using mixed design ANOVAs.

Statistical significance was set to $p < 0.05$. Analyses were performed using the GraphPad Prism 6 software (GraphPad PRISM software, version 6.01; GraphPad Software, Inc, La Jolla, CA).

**Reporting summary**. Further information on research design is available in the Nature Research Reporting Summary linked to this article.

## Data availability

All data and biological materials generated in this study are available from the corresponding author upon request. The reporting summary for this article is available as a Supplementary raw file. The source data underlying Fig. 3 are provided in Supplementary Figs. 2–6. The information of *SARDH* mutants is provided in Supplementary Fig. 9.

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

## Acknowledgements

This work was supported by grants from the Chinese Institute for Brain Research, Beijing (CIBR, Beijing) (Z181100001518004), NSFC (61625102, 61827810 and 81571044) and HTRDC (2016YFC1305903, and 2016YFC1306302), 973 Program, 2015CB856402, the Beijing Natural Science Foundation of China (Grant No. 7172022); the Scientific Research Common Program of Beijing Municipal Commission of Education (KM201510025014); and the Major Projects Fund of Beijing Institute for Brain Disorders (ZD2015-08), the startup fund of Advanced Innovation Center for Human Brain Protection (1172120205).

## Author contributions

Z.T. designed the study; L.A., T.T., Y.T., J.Y., D.C., R.W., A.W., X.F., Y.D., X.W., Y.Y., S.Z., W.W., S.B., X.Y. and R.H. performed the study; H.H., X.C., W.L. and Z.T. analyzed the results; L.A., T.T., X.C. and Z.T. wrote the paper together. All authors have read and approved the final version of the manuscript.

## Competing interests

The authors declare no competing interests.
