## [Peer Review File · Communications Biology]

Reviewers' comments:

Reviewer #1 (Remarks to the Author):

This paper entitled "Primary small molecule- formaldehyde is a memory-related molecule" investigates the role of formaldehyde in the brain in relation to its interaction with the NMDA receptor and memory learning.

The authors show that formaldehyde levels increase in the brains of rats subjected to either a learning and memory test over a 7-day period, or during hippocampal LTP induction. The increase in Formaldehyde could be suppressed by inhibitors to mitochondrial SARDH, suggesting that the increase in formaldehyde during hippocampal induction is via activation of SARDH. The authors show that direct injection of formaldehyde into the hippocampus facilitated LTP induction and memory formation. Using an inhibitor to the NMDAR, and mutagenesis strategies, they show that at low concentrations (50 microM) formaldehyde targets the C323 residue in the NR2B subunit of the NMDA receptor to facilitate LTP but at high concentrations (450microM) formaldehyde crosslinks the NR1 and NR2B subunit to reduce NMDA receptor activity.

In the mouse mutations in the ALDH gene, which degrades formaldehyde, or SARDH, which generates formaldehyde, leads to increased or decreased levels of formaldehyde, respectively. In both scenarios the learning and memory is impaired indicating that formaldehyde concentration is critical. Finally, the authors demonstrate that in Alzheimer's patients with mutations in ALDH, and in patients with pediatric sarcosinemia, and mutations in SARDH, that formaldehyde levels are altered in the same way as the mouse mutants. Finally the authors suggest that targeting the enzymes that regulate formaldehyde could be of therapeutic value.

Overall this article is interesting and tests the novel role of formaldehyde in learning and memory learning and memory. However, there are some minor points that need to be addressed prior to publication. I list these below:

1. The English still needs a lot of work, including the title and the very first sentence of the intro. In addition the word "obviously" is misleading. It suggests that this is something that the reader should already know. I think you can change this to "statistically significant" when this is the case, and/or delete the word altogether.
2. Figure 1d, did you look at FA levels in the brains of rats that were not subject to the learning and memory test? If not how do you know that these levels are not increasing just because of age?
3. Figure j-l: it is not clear in the text which neurons are being immunostained and cultured. The start of the paragraph says hippocampal neurons, and the legend says cultured neurons. I am assuming these experiments were performed on cultured hippocampal neurons, please be clear on this.
4. What are massive mitochondria? Do you mean that hippocampal neurons are highly enriched for mitochondria or that they contain mitochondria that are big in size?
5. The three medicines in this sentence need to be explained "Post hoc analyses for day 4, 5 and 6 showed that these rats injected with these three medicines had significantly less escape latency than control rats, respectively ($p < 0.01$)"
6. The links to NMDA receptor need to be explained more clearly, maybe change the order of the sentence and start with NMDA-R.
 - a. If FA could modulate NMDA-R activity 9,10, and NMDAR activity is critical for hippocampus-dependent spatial memory 11,28, thus, the antagonists of NMDA-R should have the weakening or

enhancing effect of FA on spatial memory.

TO

b. NMDAR activity is critical for hippocampus-dependent spatial memory 11,28 and FA has been shown to modulate NMDA-R activity 9,10, thus we hypothesized that antagonists of NMDA-R could have an effect of FA-associated spatial memory.

7. It would be helpful if the authors could describe the NMDA receptor before going into the mechanistic experiments.

8. Some sentences over state the finding e.g.

a. "Thus, SARDH mutation-induced FA deficiency most likely damages human cognitive functions." The data presented are correlative, and this sentence should reflect this.

9. Alzheimer's disease is a heterogenous disease, and not all patients will present with mutations in ALDH mutations. Do the authors have any insight into how broadly FA levels may affect Alzheimer's patients?

10. The statistics are mostly presented. Can the authors put p values on the escape latency graphs and details of the post-hoc test in the legend?

Reviewer #2 (Remarks to the Author):

The manuscript with the title "Primary small molecule-formaldehyde is a memory-related molecule" by Ai L, Tan T and Tang Y et al gives a detailed pathological role of formaldehyde (FA) in human diseases (AD and Sarcosinemia) mainly effecting cognitive ability. In this extensive study supported by experimental and clinical data, the authors present novel, interesting and clinically important results that can be briefed as bellow:

- LTP induction or spatial learning increases FA level. The authors use a novel technique (NaFA) to detect elevated in vivo FA levels in hippocampus, further supported by Fluo-HPLC measurement. While Glutamate increases FA production in hippocampal cell culture which is facilitated in mitochondria by Sardh, inhibiting Sardh with MA reduces FA production. Moreover, electrical stimulation in hippocampal cell culture also induces FA production. Furthermore, by increasing FA production at around 50 μ M by supplementing CT, SA or FA with intra-hippocampal injection increases LTP and spatial memory.

- In order to understand the mechanisms of FA, the authors focused on NMDA-R and show that low or high levels of FA decrease NMDA-R currents. These observations were further supported by inhibiting NR2B and NMDA-R or mutating C232A in NR2B. Inhibiting NR2B or NMDA-R decreased spatial learning and LTP in vivo further supporting the role of FA-dependent NMDA-R in memory and LTP formation. The C232A mutation shows how FA facilitates NMDA-R dependent currents. The C79 and K79 on NR1 and NR2B, respectively, were shown to be cross-linked at higher concentration of FA. As a result, NMDA-R is bi-directionally regulated by FA concentrations.

- Interestingly, low level or high level of FA has negative effect on spatial learning and LTP formation in rat hippocampus or hippocampal cell culture. The authors perform an extensive clinical study on AD patients (for high FA) and Sarcosinemia patients (for low FA. Urine FA level was higher in AD patients and hALDH2 genotyping identifies ALDH2 variants with low or complete loss of activity. Interestingly, knock-out of Aldh2 in mice, leads memory deficiency in spatial learning, which is reversed by FA injection. In order to study the deficiency of FA, the authors used clinical data from Sarcosinemia patients and identified mutations in Sardh gene, and loss of activity in blood Sardh in these patients.

Knocking-out SARDH in mice induced memory deficiency in spatial learning, further supporting the importance of FA.

The results presented here and summarized above give promises in several aspects of the functions of FA in health and pathological conditions. The experimental setup is sufficient to find the pathological role of FA and new mechanisms how FA play role in memory formation or bi-directional effect on NMDA-R.

However, there are some minor concerns and some suggestions that need to be addressed.

Minor Concerns and Suggestions

The current paper presents novel results, but it also has observations that has been published already elsewhere. Especially, spatial learning activity and LTP induction increases FA in hippocampus has been observed and published in a previous publication (Tong Z et al. 2013, Scientific Reports; PMID: 23657727). At the same, FA increase has been shown with urine analysis has already been published in a previous study (Tong Z, et al. 2009, Neurobiology of Aging; PMID: 19879019). The reviewer strongly suggests adding these references wherever the results are in accordance of previous studies. The current manuscript has already added new findings, e.g., hALDH2 variants were identified differently from previous study (Tong Z, et al. 2009, Neurobiology of Aging; PMID: 19879019). Fig S9: 3 mutations have been identified in hSARDH gene. However, there is no clear data in how many patients these mutations have been identified. Do only three patients have mutations or all of them have at least one of these mutations? If not all of them have at least one of these mutations, then how the authors explain the disease in other patients. Moreover, it is not clear how many patients have been used for sequencing. In the main text and Fig-5a, n = 11, in the material methods the number of patients is 5.

Suggestions:

The authors give some additional information that might not be required through the manuscript in parentheses. E.g., in abstract the MW of FA is given. It might be better to remove such extra information that is not required.

In page 9; C232 is written as C323.

Reviewer #3 (Remarks to the Author):

Hai and colleagues address the link between FA metabolism and cognitive disorders. They pinpoint the molecular mechanisms of dose dependent FA-regulation of NMDA-R activity and explore how FA may contribute to cognitive deficits in AD and children suffering from sarcosinemia and how FA scavengers can alleviate deficits in memory formation. The study provides helpful insights to formaldehyde dependent regulation of memory formation that has not been addressed in detail so far and can help advance the understanding of formaldehyde as a regulator of LTP in the field of neuroscience.

Essential revisions

- 1) SARDH knockout mice show a decrease in weight compared to controls. Have the authors verified that the motor abilities are not comprised compared to controls e.g. by rotarod testing? Please add this experiment to verify no impact on swimming ability.
- 2) Regarding 2b: please add data how long the FA levels stay elevated, taking measurements at later time points.

Minor revisions

- 3) For visualizing statistical significance "****" is often used. But the definition is missing in the figure legend or method section. Please comment on the identity of this abbreviation.
- 4) In Figure 1j the merged image seems to have an increase in area of the fluorescent signal. Can the authors comment on that? Is a transmitted light image superimposed? How old are the neurons in that image?
- 5) For statistical testing using t-test normal distribution has to be verified. Please provide information how this was assured.
- 6) Please consider to use a post-hoc testing of ANOVA measures if groupwise comparison was applied.
- 7) For 2f: can the authors please provide exemplary tracks of animals like in 1b?
- 8) In the former study on NMDA currents in vitro +40 mV was set as holding potential. Please comment why in the current experiments lower potentials were chosen and how this will influence the NMDA currents.
- 9) One exemplary fEPSP per condition (color coded like the fEPSP data) from the LTP measurements would be a good addition to the graph in 2d.
- 10) Supplementary figure 1: please provide an exemplary image of calcium measurements and describe how the images were processed/analyzed.

Response letter (manuscript COMMSBIO-19-0954-T)

We thank editors and reviewers and for giving us their constructive comments and suggestions which have really helped us to improve the quality of the manuscript. We have addressed these questions according to the reviewers' comments one by one. The revised sentences in revised manuscript have marked **blue**.

Reviewer #1 (Remarks to the Author):

This paper entitled "Primary small molecule- formaldehyde is a memory-related molecule" investigates the role of formaldehyde in the brain in relation to its interaction with the NMDA receptor and memory learning. The authors show that formaldehyde levels increase in the brains of rats subjected to either a test of learning and memory over a 7-day period, or during hippocampal LTP induction. The increase in Formaldehyde could be suppressed by inhibitors to mitochondrial SARDH, suggesting that the increase in formaldehyde during hippocampal induction is via activation of SARDH. The authors show that direct injection of formaldehyde into the hippocampus facilitated LTP induction and memory formation. Using an inhibitor to the NMDAR, and mutagenesis strategies, they show that at low concentrations (50 μ M) formaldehyde targets the C323 residue in the NR2B subunit of the NMDA receptor to facilitate LTP but at high concentrations (450 μ M) formaldehyde crosslinks the NR1 and NR2B subunit to reduce NMDA receptor activity.

In the mouse mutations in the ALDH2 gene, which degrade formaldehyde or SARDH, which generates formaldehyde, leads to increased or decreased levels of formaldehyde, respectively. In both scenarios the learning and memory is impaired indicating that formaldehyde concentration is critical. Finally, the authors demonstrate that in Alzheimer's patients with mutations in ALDH2, and in patients with pediatric sarcosinemia, and mutations in SARDH, that formaldehyde levels are altered in the same way as the mouse mutants. Finally the authors suggest that targeting the enzymes that regulate formaldehyde could be of therapeutic value.

Overall this article is interesting and tests the novel role of formaldehyde in learning and memory learning and memory. However, there are some minor points that need to be addressed prior to publication. I list these below:

1. The English still needs a lot of work, including the title and the very first sentence of the introduction. In addition the word "obviously" is misleading. It suggests that this is something that the reader should already know. I think you can change this to "statistically significant" when this is the case, and/or delete the word altogether.

Res: Many thanks for the reviewer's comments and suggestions. We appreciate that the reviewer has picked up language errors in title and content of the manuscript, that is very helpful for improving the quality of the manuscript. We have carefully checked and corrected these errors in the manuscript according to the reviewer's suggestion.

The previous title has been revised to “Endogenous formaldehyde is a memory-related molecule.”

2. Figure 1d, did you look at FA levels in the brains of rats that were not subject to the learning and memory test? If not how do you know that these levels are not increasing just because of age?

Res: Great question!

Recently, we examined rats’ or mice’s brain FA concentration over age. The results showed that brain FA levels were indeed increased in the aged mice and rats ¹. However, blood and brain FA levels were not statistically significant changed in the adult male mice without 7-day FA injection (**Figure 2C, 2D**) ². This result indicates that these levels are not increasing because of age.

Figure 2. Exogenous application of FA impaired LTP and spatial memory in Sprague-Dawley rats. *A, B*) Body weight (*A*) and

3. Figure j-l: it is not clear in the text which neuron is being immunostained and cultured. The start of the paragraph says hippocampal neurons, and the legend says cultured neurons. I am assuming these experiments were performed on cultured hippocampal neurons, please be clear on this.

Res: I apologized for the ambiguity.

We have revised the paragraph as the following:

“To further identify whether mitochondrial SARDH contributes to FA generation in cultured hippocampal neurons, the competitive SARDH inhibitor methoxyacetic acid (MA) was incubated before the excitatory neurotransmitter- glutamate (Glu) was added into the cultured medium.”

4. What are massive mitochondria? Do you mean that hippocampal neurons are highly enriched for mitochondria or that they contain mitochondria that are big in size?

Res: Good question.

It should be “highly enriched” and we have revised it in the section of results.

5. The three medicines in this sentence needs to be explained “Post hoc analyses for day 4, 5 and 6 showed that these rats injected with these three medicines had significantly less escape latency than control rats, respectively ($p < 0.01$)”

Res: Thanks a lot for the very helpful suggestion.

The detailed statistical data has been added in the legend of **Figure 2e** as following:

“(e) After six days of MWM training, Two-Way Repeated Measures ANOVA revealed a difference among groups: $F_{(3, 36)} = 32.43$, $p < 0.001$, time: $F_{(5, 203)} = 74.98$, $p < 0.001$, and a group/time interaction: $F_{(15, 203)} = 6.32$, $p < 0.001$. *Post hoc* tests showed that the mean escape latency values for the CT- and SA-injected group were significantly shorter than the control group in days 4, 5 and 6, $t(36) = 5.17$, $P < 0.01$; and FA-injected group had shorter escape latency than the control group in days 1, 2, 3, 4, 5 and 6, $t(36) = 7.11$, $P < 0.01$, respectively.”

6. The links to NMDA receptor needs to be explained more clearly, maybe change the order of the sentence and start with NMDA-R.

a. If FA could modulate NMDA-R activity 9,10, and NMDAR activity is critical for hippocampus-dependent spatial memory 11,28, thus, the antagonists of NMDA-R should have the weakening or enhancing effect of FA on spatial memory. TO b. NMDAR activity is critical for hippocampus-dependent spatial memory 11,28 and FA has been shown to modulate NMDA-R activity 9,10, thus we hypothesized that antagonists of NMDA-R could have an effect of FA-associated spatial memory.

Res: This is a good idea.

Following the reviewer’s suggestion, we have revised it in the section of results.

7. It would be helpful if the authors could describe the NMDA receptor before going into the mechanistic experiments.

Res: Thanks a lot for the suggestion.

We have added several sentences to describe NMDA-R as following:

“The amino acid glutamate (Glu) is recognized to be the main excitatory neurotransmitter in the brains. Glu exerts excitotoxic activity by mainly activating NMDA-type glutamate receptor (NMDAR). This receptor has been extensively studied and is the most frequently implicated in neuronal disorders³.”

8. Some sentences over state the finding e.g.

a. “Thus, SARDH mutation-induced FA deficiency most likely damages human cognitive functions.” The data presented are correlative, and this sentence should reflect this.

Res: According to this suggest, we have revised this sentence to: “Thus, SARDH mutation-induced FA deficiency may be related with human cognitive impairments.”

9. Alzheimer's disease is a heterogeneous disease, and not all patients will present with mutations in ALDH2 mutations. Do the authors have any insight into how broadly FA levels may affect Alzheimer's patients?

Res: This is an extremely interesting question.

Our previous studies showed that FA is abnormally accumulated in the brains of AD animal models and AD patients⁵⁹. Additionally, the brain aging process decreases formaldehyde dehydrogenase (FDH, another formaldehyde degrading enzyme) expression and gradually inactivates the enzyme associated with FA accumulation^{33,34}. FA not only directly induces mitochondrial toxicity^{22,60}, but also promotes A β endocytosis to form oligomers and synergistically enhances A β mitochondrial toxicity. More importantly, FA directly induces Tau hyperphosphorylation, which results in the formation of neuronal fibril tangles (NTFs)⁶¹ and spatial memory deficits in wild-type mice¹⁵. Knockout of FDH results in defective visual memory in *Drosophila*⁶². These data strongly suggest that age-related FA directly impairs cognitive functions in the aged people and accelerates cognitive loss in AD patients.

10. The statistics are mostly presented. Can the authors put p values on the escape latency graphs and details of the post-hoc test in the legend?

Res: Thanks a lot for the suggestion.

We have added *p* values in the graphs of Fig. 2e, 2g, Fig. 4b, 4e, Fig. 5g and Fig. S11a.

These statistical data of the *Tukey post-hoc* test were added in the legends of figure 2 and figure 4.

Reviewer #2 (Remarks to the Author):

The manuscript with the title "Primary small molecule-formaldehyde is a memory-related molecule" by Ai L, Tan T and Tang Y et al gives a detailed pathological role of formaldehyde (FA) in human diseases (AD and Sarcosinemia) mainly effecting cognitive ability. In this extensive study supported by experimental and clinical data, the authors present novel, interesting and clinically important results that can be briefed as bellow:

LTP induction or spatial learning increases FA level. The authors use a novel technique (NaFA) to detect elevated in vivo FA levels in hippocampus, further supported by Fluo-HPLC measurement. While Glutamate increases FA production in hippocampal cell culture which is facilitated in mitochondria by Sardh, inhibiting Sardh with MA reduces FA production. Moreover, electrical stimulation in hippocampal cell culture also induces FA production. Furthermore, by increasing FA production at around 50 μ M by supplementing CT, SA or FA with intra-hippocampal injection increases LTP and spatial memory.

In order to understand the mechanisms of FA, the authors focused on NMDA-R and show that low or high levels of FA decrease NMDA-R currents. These observations were further supported by inhibiting NR2B and NMDA-R or mutating C232A in

NR2B. Inhibiting NR2B or NMDA-R decreased spatial learning and LTP in vivo further supporting the role of FA-dependent NMDA-R in memory and LTP formation. The C232A mutation shows how FA facilitates NMDA-R dependent currents. The C79 and K79 on NR1 and NR2B, respectively, were shown to be cross-linked at higher concentration of FA. As a result, NMDA-R is bi-directionally regulated by FA concentrations.

Interestingly, low level or high level of FA has negative effect on spatial learning and LTP formation in rat hippocampus or hippocampal cell culture. The authors perform an extensive clinical study on AD patients (for high FA) and Sarcosinemia patients (for low FA. Urine FA level was higher in AD patients and hALDH2 genotyping identifies ALDH2 variants with low or complete loss of activity. Interestingly, knock-out of Aldh2 in mice, leads memory deficiency in spatial learning, which is reversed by FA injection. In order to study the deficiency of FA, the authors used clinical data from Sarcosinemia patients and identified mutations in Sardh gene, and loss of activity in blood Sardh in these patients. Knocking-out Sardh in mice induced memory deficiency in spatial learning, further supporting the importance of FA.

The results presented here and summarized above give promises in several aspects of the functions of FA in health and pathological conditions. The experimental setup is sufficient to find the pathological role of FA and new mechanisms how FA play role in memory formation or bi-directional effect on NMDA-R.

Res: Many thanks for the reviewer's comments.

However, there are some minor concerns and some suggestions that need to be addressed.

Minor Concerns and Suggestions

The current paper presents novel results, but it also has observations that has been published already elsewhere. Especially, spatial learning activity and LTP induction increases FA in hippocampus has been observed and published in a previous publication (Tong Z et al. 2013, Scientific Reports; PMID: 23657727). At the same, FA increase has been shown with urine analysis has already been published in a previous study (Tong Z, et al. 2009, Neurobiology of Aging; PMID: 19879019). The current manuscript has already added new findings, e.g., hALDH2 variants were identified differently from previous study (Tong Z, et al. 2009, Neurobiology of Aging; PMID: 19879019). The reviewer strongly suggests adding these references wherever the results are in accordance of previous studies.

Res: We apologized for that we did not accurately cite our previous studies.

We have added these references in revised manuscript.

In this study, 50 μ M FA-induced enhancement of LTP and memory served as a positive control to compare with the effects of FA precursor- SA (sarcosine) and CT (creatine). Hence, the experiment of 50 μ M FA treatment was necessary to be repeated (**Fig. 2d, 2e**).

In addition, as the review said, this present study was based on our previous researches and further provided some new evidences, such as: ALDH2 variants

“Our previous study showed that urine FA levels were negatively correlated with cognitive function in AD patients ⁸. To establish the relationship between FA overload and ALDH2 activity (a FA-degrading enzyme) or cognitive decline, the cognitive abilities of 158 participants were examined using the Mini-Mental State Examination (MMSE ⁸) together with analysis the genotype of *ALDH2* and urinalysis of FA (Table S1).”

Fig S9: 3 mutations have been identified in *hSARDH* gene. However, there is no clear data in how many patients these mutations have been identified. Do only three patients have mutations or all of them have at least one of these mutations? If not all of them have at least one of these mutations, then how the authors explain the disease in other patients. Moreover, it is not clear how many patients have been used for sequencing. In the main text and Fig-5a, n = 11, in the material methods the number of patients is 5.

Res: Many thanks for the helpful comments.

We found that the three mutations of *hSARDH* gene were distributed in these 11 sarcosinemia children associated with severe low activity of SARDH and cognitive decline.

Our double-blind experiment also showed other mutations of *hSARDH* gene by high-throughput gene sequencing in 42 children. However, whether these mutations are related with SARDH activity and cognitive functions needs to be further investigated.

Yes, it was an error. The numbers of preschool children with sarcosinemia in Supplementary materials should be 11 as showed in **Fig. 5a**. We have revised it.

Suggestions:

The authors give some additional information that might not be required through the manuscript in parentheses. E.g., in abstract the MW of FA is given. It might be better to remove such extra information that is not required.

Res: Thanks a lot for the suggestions. We have deleted the information in revised manuscript.

In page 9; C232 is written as C323.

Res: Yes, it has been corrected to C232 in revised manuscript.

Reviewer #3 (Remarks to the Author):

Li Ai and colleagues address the link between FA metabolism and cognitive disorders. They pinpoint the molecular mechanisms of dose dependent FA-regulation of NMDA-R activity and explore how FA may contribute to cognitive deficits in AD and children suffering from sarcosinemia and how FA scavengers can alleviate deficits in

memory formation.

The study provides helpful insights to formaldehyde dependent regulation of memory formation that has not been addressed in detail so far and can help advance the understanding of formaldehyde as a regulator of LTP in the field of neuroscience.

Essential revisions

1) SARDH knockout mice show a decrease in weight compared to controls. Have the authors verified that the motor abilities are not comprised compared to controls e.g. by rotarod testing? Please add this experiment to verify no impact on swimming ability.

Res: Many thanks for the reviewer's comments.

According to the reviewer's suggestion, we have added the accelerating rotarod test in revised manuscript (**Supplementary Fig. 10e, 10f**) to compare the motor abilities between wild-type control mice and *SARDH*^{-/-} mice. The test shows that there is no significant difference in motor ability between the two groups. The following is the new supplementary figure 10.

2) Regarding 2b: please add data how long the FA levels stay elevated, taking measurements at later time points.

Res: Good suggestion!

We found that injection of FA, CT and SA induced a marked elevation in brain FA

levels at 30 minutes and then gradually declined to the baseline levels (n=6, per time point/per group) as showed in the following figure.

Minor revisions

3) For visualizing statistical significance “****” is often used. But the definition is missing in the figure legend or method section. Please comment on the identity of this abbreviation.

Res: Thanks a lot for the suggestion!

We have added the *p* values on the graphs of Fig. 2e, 2g, Fig. 4b, 4e, Fig. 5g and Fig. S11a.

These statistical data of the *Tukey post-hoc* test were added in the Figure legends.

4) In Figure 1j the merged image seems to have an increase in area of the fluorescent signal. Can the authors comment on that? Is a transmitted light image superimposed? How old are the neurons in that image?

Res: We agree with the reviewer’s comments. In order to make it easier to identify the co-localization of SARDH (red) and Cox IV (green, a mitochondrial marker), the merged image was superimposed.

The imaged hippocampal neurons used in this experiment had been cultured for at least two weeks for the formation of synaptic connections.

5) For statistical testing using t-test normal distribution has to be verified. Please provide information how this was assured.

Res: In the revised manuscript, we have added the statistical methods in the section of Supplementary Materials as following:

“All data were tested for normality by the *Kolmogorov-Smirnov* test. When data were normally distributed, the statistical significance of differences was assessed with the unpaired *t* test and one or two-way ANOVA, analyzed by *Tukey post hoc*. When data were not normally distributed, the statistical significance of differences was judged on the basis of *P* values with the *Mann-Whitney U* test and analyzed by IBM SPSS v19.0 (SPSS Inc., Chicago, IL, USA).

Human serum biochemical index were assessed using Student's unpaired *t*-test. Gender of participants was assessed using the *chi*-squared test.

Correlations between urine FA levels and MMSE or WISC scores were assessed using *Pearson* correlation coefficient, both without adjustment, then accounting for sex and age. These data were analyzed by IBM SPSS v19.0.

Mice serum biochemical index were assessed using Student's unpaired *t*-test. The spatial memory behaviors in Morris water maze of mice were analyzed by *Tukey post hoc* with repeated measures ANOVA.

The changes in the response amplitudes of LTP were analyzed using mixed design ANOVAs.

Statistical significance was set to $p < 0.05$. Analyses were performed using the GraphPad Prism 6 software (GraphPad PRISM software, version 6.01; GraphPad Software, Inc, La Jolla, CA).”

6) Please consider to use a post-hoc testing of ANOVA measures if group wise comparison was applied.

Res: Many thanks for the suggestion.

The spatial memory behaviors in Morris water maze of mice were analyzed by *Tukey post hoc* with repeated measures ANOVA.

The described statistical data have been added in the section of results and Figure legends.

7) For 2f: can the authors please provide exemplary tracks of animals like in 1b?

Res: Good suggestion! We have added the swimming tracks in the **Figure 2f** (Top).

8) In the former study on NMDA currents in vitro +40 mV was set as holding potential. Please comment why in the current experiments lower potentials were chosen and how this will influence the NMDA currents.

Res: In the former study, cell membrane potential was depolarized to +40 mV to reduce of voltage-dependent Mg^{2+} blockade.

In the current study, magnesium free external solution was applied to record NMDA current at -60 mV, which is close to rest membrane potential.

9) One exemplary fEPSP per condition (color coded like the fEPSP data) from the LTP measurements would be a good addition to the graph in 2d.

Res: Good idea! We have added it in the **Figure 2d (Top)**.

10) Supplementary figure 1: please provide an exemplary image of calcium measurements and describe how the images were processed or analyzed.

Res: Thanks a lot for the suggestion. According to the suggestion, we have included the method of calcium imaging the section of supplementary Materials as following: r

“Intracellular Ca^{2+} imaged by laser confocal

The $[Ca^{2+}]_i$ imaging in the CHO cells was used with Fluo-3 of Ca^{2+} probe. Another Ca^{2+} probe- Fluo-4 was used to the $[Ca^{2+}]_i$ imaging in the cultured hippocampal neurons described as previously¹⁸.”

The $[Ca^{2+}]_i$ influx imaging with Fluo-4 probe was described as the following **Figures**:

1. The baseline of $[Ca^{2+}]_i$ was recorded for 15 seconds.
2. Then the agonist-NMDA or the mixed solutions of Ifen and NMDA was

added into the medium of cultured CHO cells; and the dynamic changes in $[Ca^{2+}]_i$ were real-time recorded for at least 200 seconds.

3. After the recording process of $[Ca^{2+}]_i$ was finished, several cells in the visual field were selected for statistical analysis of the Ca^{2+} -fluorescence intensity.

Reference:

- 1 Tong Z, Han C, Luo W, Wang X, Li H, Luo H, Zhou J, Qi J, He R. Accumulated hippocampal formaldehyde induces age-dependent memory decline. *Age (Dordr)* **35**, 583-596, (2013).
- 2 Tan, T. *et al.* Formaldehyde induces diabetes-associated cognitive impairments. *FASEB J*, fj201701239R, (2018).
- 3 Monyer, H. *et al.* Heteromeric NMDA receptors: molecular and functional distinction of subtypes. *Science* **256**, 1217-1221, (1992).